# A UNIFIED PERSPECTIVE ON FINE-TUNING AND SAMPLING WITH DIFFUSION AND FLOW MODELS

## ABSTRACT

We study the problem of training diffusion and flow generative models to sample from target distributions defined by an exponential tilting of the base density. This tasks subsumes sampling from unnormalized densities and reward fine-tuning a pre-trained model, and can be approached from a stochastic optimal control (SOC) perspective and from a thermodynamics perspective. The SOC formulation has been tackled using adjoint-based methods (Adjoint Matching and Sampling), and score matching methods, while the thermodynamics formulation has given rise to algorithms such as CMCD and NETS. Our contributions include bounding the lean adjoint ODE underlying Adjoint Matching and Sampling, deriving bias–variance decompositions that allow a principled comparison between adjoint-based and score-matching methods, adapting thermodynamic formulations to the exponential tilting setting, and text-to-image fine-tuning experiments.

## 1 INTRODUCTION

Recent advances in generative modeling have demonstrated the effectiveness of diffusion and flow matching models for learning complex data distributions (Song et al., 2021; Ho et al., 2020; Lipman et al., 2022; Albergo et al., 2023; Liu et al., 2023). In many applications, however, it is desirable to tailor the generative process to favor certain qualities—either by sampling from an unnormalized target distribution or by fine-tuning a pre-trained model with a reward function (Uehara et al., 2024; Domingo-Enrich et al., 2025; Zhang & Chen, 2022; Holdijk et al., 2023). A natural way to address these challenges is via *exponential tilting*, wherein a base density is modified to a target one by reweighting with an exponential factor. This formulation not only unifies reward fine-tuning and sampling from unnormalized distributions but also naturally lends itself to analysis using tools from stochastic optimal control (SOC), partial differential equation (PDE) analysis, stochastic processes and thermodynamics.

Motivated by the broad applications, such as text-to-image generation (Domingo-Enrich et al., 2025) and protein design (Wang et al., 2025), a growing body of work has explored methods for fine-tuning diffusion and flow models. Although the underlying problem structure aligns well with SOC, particularly when casting as controlling an SDE with a reward function, the key challenge is the bias introduced by lifting the base process with an additional control. Uehara et al. (2024) proposed a two-step strategy with an additional control to draw samples from the biased initial distribution. Adjoint matching (Domingo-Enrich et al., 2025) systematically analyzed this problem and introduced memoryless noise schedules to solve this issue. More recently, reinforcement learning (RL)-based approaches have also been introduced, e.g. GRPO (Liu et al., 2025b).

In a parallel literature, training diffusion and flow models to sample from unnormalized densities has also been largely studied. Early work (Zhang & Chen, 2022; Vargas et al., 2023; Berner et al., 2022) can be viewed as solving an SOC problem where the reward function is related to the unnormalized density. Another branch of work (Phillips et al., 2024; Akhound-Sadegh et al., 2024) leveraged the target score identity, which relates the score to the unnormalized target density and learns an SDE to draw samples. The idea of adjoint matching has also been applied to this problem, together with reciprocal projection for efficient off-policy training (Havens et al., 2025; Liu et al., 2025a). More recently, Vargas et al. (2024) and Albergo & Vanden-Eijnden (2025) introduced a thermodynamic formulation of this problem, extending the classical annealed importance sampling (AIS) and Jarzynski equality perspective using path-space variational inference and PDE formulations.

In this paper, we provide a unifying framework that formulates training diffusion and flow models to sample from an exponentially tilted target distribution, which covers reward-based fine-tuning of a pretrained model, as well as sampling from unnormalized densities. Our main contributions are as follows: *(i)* we bound the norm of the lean adjoint ODE for Adjoint Matching and Sampling (AM/AS), supporting the empirical performance of the algorithms; *(ii)* we derive bias-variance decompositions for adjoint-based and score matching algorithms to compare the algorithms on equal footing, under which AM/AS and Novel Score Matching (NSM, Phillips et al. (2024)) perform favorably; *(iii)* we adapt the thermodynamic framework of Vargas et al. (2024) and Albergo & Vanden-Eijnden (2025) to the exponential tilting problem which yields analogs of the Controlled Monte Carlo Diffusions (CMCD) and Non-Equilibrium Transport Sampler (NETS) loss functions, as well as novel variants of the celebrated Crooks fluctuation theorem and Jarzynski equality. Finally, we perform reward-based fine-tuning experiments on Stable Diffusion 1.5 and 3 using Adjoint Matching, refining the techniques of Domingo-Enrich et al. (2025).

## 2 BACKGROUND

### 2.1 FLOW MATCHING, DDIM AND FÖLLMER PROCESSES

**Flow Matching or Stochastic Interpolants** Given a real-valued differentiable function $(\alpha_t)_{t\in[0,1]}$ such that $\alpha_0 = 0$, $\alpha_1 = 1$, and a real-valued differentiable function $(\beta_t)_{t\in[0,1]}$ such that $\beta_1 = 0$, the reference flow or stochastic interpolant $\bar{X}$ is defined as

$$\bar{X}_t = \alpha_t Y + \beta_t \varepsilon, \tag{1}$$

where $\varepsilon \sim p_0 = \mathcal{N}(0, \mathrm{I})$, $Y \sim p_{\mathrm{data}}$. If we let

$$\kappa_t = \frac{\dot{\alpha}_t}{\alpha_t}, \qquad \eta_t = \beta_t\big(\frac{\dot{\alpha}_t}{\alpha_t}\beta_t - \dot{\beta}_t\big), \tag{2}$$

the Flow Matching reference SDE, whose solution has the same time marginals as the reference flow up to a time flip, is

$$d\tilde{X}_t = -\kappa_{1-t}\tilde{X}_t \, dt + \sqrt{2\eta_{1-t}} \, dB_t, \qquad \tilde{X}_0 \sim p_{\mathrm{data}}. \tag{3}$$

That is, for all $t \in [0, 1]$, $\vec{X}_t$ and $\bar{X}_{1-t}$ have the same distribution. And the optimal generative SDE reads

$$dX_t = \big(\kappa_t X_t + \big(\frac{\sigma(t)^2}{2} + \eta_t\big)\mathfrak{s}_t(X_t)\big) \, dt + \sigma(t) \, dB_t, \qquad X_0 \sim \mathcal{N}(0, \beta_0^2 \mathrm{I}), \tag{4}$$

where $\mathfrak{s}_t$ is the score function of the reference process: $\mathfrak{s}_t(x) = \nabla \log p_t(x)$, and the noise schedule $\sigma$ is arbitrary. For any $\sigma$, the marginals $X_t$ also have the same distribution as the reference flow marginals $\bar{X}_t$. Setting $\sigma(t) = \sqrt{2\eta_t}$ yields the memoryless process, which is the time reversal of the process in (3):

$$dX_t = \big(\kappa_t X_t + 2\eta_t \mathfrak{s}_t(X_t)\big) \, dt + \sqrt{2\eta_t} \, dB_t, \qquad X_0 \sim \mathcal{N}(0, \beta_0^2 \mathrm{I}). \tag{5}$$

**Remark 2.1.** *Although the drift terms in (4) and (5) are written in terms of the score to facilitate the analysis, in Flow Matching the vector field that is learned is $v_t(x) = \kappa_t X_t + \eta_t \mathfrak{s}_t(x)$, and in DDIM it is often the denoiser $\epsilon_t(x) = -\beta_t \mathfrak{s}_t(x)$. These vector fields encode the same information, and the drifts can be written in terms of any of them.*

**Subcases of Flow Matching** The following processes are particular instances of flow matching:

(i) *Föllmer process:* Given an increasing function $\bar{\alpha}_t$ such that $\bar{\alpha}_0 = 0$ and $\bar{\alpha}_1 = 1$, and a constant $\sigma_0^2 > 0$, the interpolant $\bar{X}$ and the coefficients $\alpha_t$, $\beta_t$, $\kappa_t$ and $\eta_t$ read

$$\bar{X}_t = \bar{\alpha}_t Y + \sqrt{\bar{\alpha}_t(1-\bar{\alpha}_t)}\sigma_0\varepsilon \implies \alpha_t = \bar{\alpha}_t, \quad \beta_t = \sqrt{\bar{\alpha}_t(1-\bar{\alpha}_t)}\sigma_0,$$
$$\dot{\alpha}_t = \dot{\bar{\alpha}}_t, \quad \dot{\beta}_t = \frac{(1-2\bar{\alpha}_t)\dot{\bar{\alpha}}_t}{2\sqrt{\bar{\alpha}_t(1-\bar{\alpha}_t)}}\sigma_0^2, \implies \kappa_t = \frac{\dot{\bar{\alpha}}_t}{\bar{\alpha}_t}, \quad \eta_t = \frac{\dot{\bar{\alpha}}_t\sigma_0^2}{2}. \tag{6}$$

(ii) *DDIM/DDPM:* Given an increasing function $\bar{\alpha}_t$ such that $\bar{\alpha}_0 = 0$ and $\bar{\alpha}_1 = 1$, and a constant $\sigma_0^2 > 0$, the interpolants and coefficients $\alpha_t$, $\beta_t$, $\kappa_t$ and $\eta_t$ read

$$\bar{X}_t = \sqrt{\bar{\alpha}_t}Y + \sqrt{1-\bar{\alpha}_t}\sigma_0\varepsilon \implies \alpha_t = \sqrt{\bar{\alpha}_t}, \quad \beta_t = \sqrt{1-\bar{\alpha}_t}\sigma_0,$$
$$\dot{\alpha}_t = \frac{\dot{\bar{\alpha}}_t}{2\sqrt{\bar{\alpha}_t}}, \quad \dot{\beta}_t = -\frac{\dot{\bar{\alpha}}_t}{2\sqrt{1-\bar{\alpha}_t}}\sigma_0, \implies \kappa_t = \frac{\dot{\bar{\alpha}}_t}{2\bar{\alpha}_t}, \quad \eta_t = \frac{\dot{\bar{\alpha}}_t\sigma_0^2}{2\bar{\alpha}_t}. \tag{7}$$

(iii) *Rectified Flow (OT Flow Matching):* Given an increasing function $\bar{\alpha}_t$ such that $\bar{\alpha}_0 = 0$ and $\bar{\alpha}_1 = 1$, and a constant $\sigma_0^2 > 0$, the interpolant $\bar{X}$ and the coefficients $\alpha_t$, $\beta_t$, $\kappa_t$ and $\eta_t$ read

$$\bar{X}_t = \bar{\alpha}_t Y + (1 - \bar{\alpha}_t)\sigma_0 \varepsilon \implies \alpha_t = \bar{\alpha}_t, \quad \beta_t = (1 - \bar{\alpha}_t)\sigma_0.$$

$$\dot{\alpha}_t = \dot{\bar{\alpha}}_t, \quad \dot{\beta}_t = -\dot{\bar{\alpha}}_t, \implies \kappa_t = \frac{\dot{\bar{\alpha}}_t}{\bar{\alpha}_t}, \quad \eta_t = \frac{(1-\bar{\alpha}_t)\dot{\bar{\alpha}}_t \sigma_0^2}{\bar{\alpha}_t}. \tag{8}$$

## 2.2 EXPONENTIAL TILTING AND ITS CONTROL AND THERMODYNAMIC FORMULATIONS

**The exponential tilting problem** Given a Flow Matching model that generates a distribution $p^{\text{base}}$ over $\mathbb{R}^d$ and a function $r : \mathbb{R}^d \to \mathbb{R}$, consider the task of modifying the model such that the generated distribution is the following tilted distribution:

$$p^{\star}(x) \propto p^{\text{base}}(x) \exp(r(x)). \tag{9}$$

Two main settings are covered by the exponential tilting framework:

(i) *Reward fine-tuning:* $p^{\text{base}}$ is the distribution generated by a pre-trained diffusion or flow matching model, and $r$ is a reward model that takes high values for high quality samples.

(ii) *Sampling from unnormalized distributions:* The goal is to sample from the unnormalized distribution proportional to $p^{\star}(x) = \exp(-E(x))$, where $E$ is the energy function. $p^{\text{base}}$ is a Gaussian $\mathcal{N}(0, \sigma_1^2 \mathrm{I})$, and the reward $r$ is chosen to be $r(x) = -E(x) + \frac{\|x\|^2}{2\sigma_1^2}$.

While the Föllmer process is typically used for sampling (e.g. Havens et al. (2025)), and DDIM and OT Flow Matching are commonly used for generative modeling, the choice of process is independent of the application.

**The stochastic optimal control formulation** Domingo-Enrich et al. (2025) proves that the exponential tilting problem can be reformulated as the following stochastic optimal control (SOC) problem:[1]

$$\min_{u \in \mathcal{U}} \mathbb{E}\left[\frac{1}{2} \int_0^1 \|u(X_t^u, t)\|^2 \, \mathrm{d}t + g(X_1^u)\right], \tag{10}$$

$$\text{s.t. } \mathrm{d}X_t^u = (b_\sigma(X_t^u, t) + \sigma(t)u(X_t^u, t)) \, \mathrm{d}t + \sigma(t)\mathrm{d}B_t, \qquad X_0^u \sim p_0 = \mathcal{N}(0, \beta_0^2 \mathrm{I}), \tag{11}$$

$$\text{where} \quad b_\sigma(x, t) = \kappa_t x + \left(\frac{\sigma(t)^2}{2} + \eta_t\right)\mathfrak{s}_t(x), \tag{12}$$

as long as the uncontrolled process $X = X^0$ satisfies the memoryless property, meaning that $X_0$ and $X_1$ are statistically independent. (Domingo-Enrich et al., 2025, Prop. 1) shows that a generative process is memoryless if and only if the noise schedule is chosen as $\sigma(t)^2 = 2\eta_t + \chi(t)$, with $\chi : [0, 1] \to \mathbb{R}$ such that for all $t \in (0, 1]$, $\lim_{t' \to 0^+} \alpha_{t'} \exp\left(-\int_{t'}^t \frac{\chi(s)}{2\beta_s^2} \, \mathrm{d}s\right) = 0$. In particular, they refer to $\sigma(t) = \sqrt{2\eta_t}$ as *the* memoryless noise schedule, as it is the only one such that the resulting fine-tuned model can be used to perform inference with an arbitrary noise schedule (Domingo-Enrich et al., 2025, Thm. 1).

**The thermodynamics formulation** Methods like CMCD (Vargas et al., 2024) and NETS (Albergo & Vanden-Eijnden, 2025) were developed in a setting where one has access to a time-dependent energy function $(U_t)_{t \in [0,1]}$, instead of a flow matching vector field that generates $p_{\text{base}}$ and a time-dependent reward function $(r_t)_{t \in [0,1]}$. That is, their goal is to learn a vector field that yields a process with marginals $p_t^\star(x) \propto \exp(-U_t(x))$. This is different but related to the task of learning a vector field that yields a process with marginals $p_t^\star(x) \propto p_t^{\text{base}}(x)\exp(r_t(x))$, which solves the exponential tilting problem (9) if $(r_t)_{t \in [0,1]}$ is chosen such that $r_1 = r$ and $p_0^\star \propto \mathcal{N}(0, \beta_0^2)\exp(r_0)$ is easy to sample from. More specifically, we consider processes $X^v$ of the form

$$\mathrm{d}X_t^v = (b_\sigma^r(X_t^v, t) + v(X_t^v, t)) \, \mathrm{d}t + \sigma(t)\mathrm{d}B_t, \quad \begin{cases} X_0^v \sim p_0^\star \propto \mathcal{N}(0, \beta_0^2 \mathrm{I})\exp(r_0), \\ b_\sigma^r(x, t) := \kappa_t x + \left(\frac{\sigma(t)^2}{2} + \eta_t\right)\left(\mathfrak{s}_t(x) + \nabla r_t(x)\right). \end{cases} \tag{13}$$

---

[1]In (12), $\mathfrak{s}_t$ is the score function corresponding to $p^{\text{base}}$. In practice, $b_\sigma$ is computed using the learned FM vector field or denoiser (Rem. 2.1).

and our goal is to learn a $v^\star$ such that the marginal distribution of $X_t^{v^\star}$ is $p_t^\star(x) \propto p_t^{\text{base}}(x) \exp(r_t(x))$. Unlike in the SOC formulation, in the thermodynamics formulation the noise schedule $\sigma(t)$ can be picked arbitrarily.

## 3 STOCHASTIC OPTIMAL CONTROL ALGORITHMS AND ANALYSIS

### 3.1 ADJOINT-BASED METHODS

**Adjoint Matching** Adjoint Matching (AM) is a SOC deep learning loss introduced by Domingo-Enrich et al. (2025). For the exponential tilting problem with scalar $\sigma$, it takes the following form:

$$\mathcal{L}_{\text{Adj-Match}}(u; X^{\bar{u}}) := \tfrac{1}{2} \int_0^1 \left\| u(X_t^{\bar{u}}, t) + \sigma(t)\tilde{a}(t; X^{\bar{u}}) \right\|^2 \mathrm{d}t, \qquad \bar{u} = \texttt{stopgrad}(u), \quad (14)$$

$$\text{where} \quad \tfrac{\mathrm{d}}{\mathrm{d}t} \tilde{a}(t; X^{\bar{u}}) = -\nabla_x b_\sigma(X_t^{\bar{u}}, t)^\top \tilde{a}(t; X^{\bar{u}}), \quad (15)$$

$$\tilde{a}(1; X^{\bar{u}}) = -\nabla_x r(X_1^{\bar{u}}). \quad (16)$$

Domingo-Enrich et al. (2025) refer to the ODE (15)-(16) as the lean adjoint ODE. They show that if $u$ is a critical point of the expected loss $\mathcal{L}_{\text{Adj-Match}}(u) := \mathbb{E}[\mathcal{L}_{\text{Adj-Match}}(u; X^{\bar{u}})]$ that is, if $\frac{\delta}{\delta u} \mathcal{L}_{\text{Adj-Match}}(u) \equiv 0$, then $u$ is the optimal control $u^\star$.

**Adjoint Sampling** Adjoint Sampling was introduced by Havens et al. (2025) as a procedure based on Adjoint Matching to sample from unnormalized densities, with improved efficiency. When $p^{\text{base}}$ is a Gaussian $\mathcal{N}(0, \sigma_1^2 I)$, we have that $\bar{X}_t \sim \mathcal{N}(0, (\alpha_t^2 \sigma_1^2 + \beta_t^2) I)$, which means that $b(x, t)$ defined in (12) is a linear function of $x$: $b_\sigma(x, t) = \left( \kappa_t - \frac{\sigma(t)^2 + 2\eta_t}{2(\alpha_t^2 \sigma_1^2 + \beta_t^2)} \right) x := \chi_t x$. Hence, the adjoint ODE (15) needs not be solved as it admits a closed form solution: $\tilde{a}(t; X^{\bar{u}}) = -\exp\left( \int_t^1 \chi_s \, \mathrm{d}s \right) \nabla_x r(X_1^{\bar{u}})$. To obtain a further speed-up, in Adjoint Sampling the loss is not evaluated at intermediate points of the trajectory $X^{\bar{u}}$, but rather at points $\bar{X}_t^{\bar{u}} = \alpha_t X_1^{\bar{u}} + \beta_t \varepsilon$ obtained by noising the final iterate $X_1^{\bar{u}}$:

$$\mathcal{L}_{\text{Adj-Sampl}}(u) := \mathbb{E}_{\bar{X}_t^{\bar{u}} = \alpha_t X_1^{\bar{u}} + \beta_t \varepsilon} \left[ \tfrac{1}{2} \int_0^1 \left\| u(\bar{X}_t^{\bar{u}}, t) - \exp\left( \int_t^1 \chi_s \, \mathrm{d}s \right) \sigma(t) \nabla_x r(X_1^{\bar{u}}) \right\|^2 \mathrm{d}t \right] \quad (17)$$

Each rollout $X_1^{\bar{u}}$ can be noised multiple times, which yields an algorithm which is much more efficient than AM, even though it is more restrictive because it only works for sampling.

The behavior of Adjoint Matching and Sampling depeds heavily on the norm of the solution to the lean adjoint ODE. In the following proposition we bound the norm of $\tilde{a}(t; X^{\bar{u}})$ under convexity assumptions on $p^{\text{base}}$.

**Proposition 3.1** (Norm of the lean adjoint state). *Let $\tilde{a}(t, X^u)$ be the solution of the lean adjoint ODE* (15)-(16) *with memoryless schedule $\sigma(t) = \sqrt{2\eta_t}$. Assume there exists $\sigma_1^2 > 0$ such that the density $p^{\text{base}}$ is $\frac{1}{\sigma_1^2}$-strongly log-concave, i.e. $-\nabla^2 \log p^{\text{base}}(x) \succeq \frac{1}{\sigma_1^2} I$. Let $\chi_t := \kappa_t - \frac{2\eta_t}{\beta_t^2 + \alpha_t^2 \sigma_1^2}$. For all $t \in [0, 1]$, we have that*

$$\|\tilde{a}(t, X^u)\| \leq \exp\left( \int_t^1 \chi_s \, \mathrm{d}s \right) \|\nabla_x r(X_1^u)\|. \quad (18)$$

*In particular,* (i) *for the Föllmer schedule,* $\exp\left( \int_t^1 \chi_s \, \mathrm{d}s \right) = \frac{\sigma_1^2}{(1 - \bar{\alpha}_t)\sigma_0^2 + \bar{\alpha}_t \sigma_1^2}$, (i) *for DDPM/DDIM,* $\exp\left( \int_t^1 \chi_s \, \mathrm{d}s \right) = \frac{\sigma_1^2 \sqrt{\bar{\alpha}_t}}{(1 - \bar{\alpha}_t)\sigma_0^2 + \bar{\alpha}_t \sigma_1^2}$, *and* (i) *for DDPM/DDIM,* $\exp\left( \int_t^1 \chi_s \, \mathrm{d}s \right) = \frac{\bar{\alpha}_t}{(1 - \bar{\alpha}_t)^2 + \bar{\alpha}_t^2}$. *Moreover, when $p^{\text{base}} = \mathcal{N}(0, \sigma_1^2 I)$ as in Adjoint Sampling, equation (18) holds with equality.*

The proof of Prop. 3.1 in App. B.1 relies on bounding the spectrum of $\text{Sym}(\nabla b_\sigma)$, which is connected to the spectrum of $\nabla^2 \log p^{\text{base}}$. While strong convexity of $p^{\text{base}}$ is a strong condition in the fine-tuning case, since spectra are local properties, as long as the trajectory $X^{\bar{u}}$ spends most of the time $t \in [0, 1]$ in regions where $p_t^{\text{base}}$ is locally strongly log-concave (basins of $p_t^{\text{base}}$), the norm $\|\tilde{a}(t, X^{\bar{u}})\|$ will decay accordingly. This explains the norm decay and consequent good performance of the algorithm in realistic settings (Sec. 5).

### 3.2 SCORE MATCHING METHODS

In what follows, we include a review of existing score-based generative modeling methods (see derivations in App. B.2). These methods are designed to learn arbitrary distributions $p_{\text{data}}$; and all

require having access to samples from $p_{\text{data}}$ as well as the noiseless score $\nabla \log p_{\text{data}}$, except for Conditional Score Matching, which only relies on samples from $p_{\text{data}}$. These loss functions follow from these three identities involving the score function of the distribution of $\bar{X}_t$ (see Thm. A.1):

*Conditional score identity:*
$$\nabla \log p_t(x) = -\frac{\int_{\mathbb{R}^d} (x - \alpha_t y) \mathcal{N}(x; \alpha_t y, \beta_t^2 \mathrm{I}) \, p_{\text{data}}(y) \, \mathrm{d}y}{\beta_t^2 \int_{\mathbb{R}^d} \mathcal{N}(x; \alpha_t y, \beta_t^2 \mathrm{I}) \, p_{\text{data}}(y) \, \mathrm{d}y}, \tag{CSI}$$

*Target score identity:*
$$\nabla \log p_t(x) = \frac{\int_{\mathbb{R}^d} \nabla \log p_{\text{data}}(y) \mathcal{N}(x; \alpha_t y, \beta_t^2 \mathrm{I}) p_{\text{data}}(y) \, \mathrm{d}y}{\alpha_t \int_{\mathbb{R}^d} \mathcal{N}(x; \alpha_t y, \beta_t^2 \mathrm{I}) p_{\text{data}}(y) \, \mathrm{d}y}. \tag{TSI}$$

*Novel score identity:*
$$\nabla \log p_t(x) = \frac{\int_{\mathbb{R}^d} (\alpha_t \nabla \log p_{\text{data}}(y) - (x - \alpha_t y)) \mathcal{N}(x; \alpha_t y, \beta_t^2 \mathrm{I}) \, p_{\text{data}}(y) \, \mathrm{d}y}{(\alpha_t^2 + \beta_t^2) \int_{\mathbb{R}^d} \mathcal{N}(x; \alpha_t y, \beta_t^2 \mathrm{I}) \, p_{\text{data}}(y) \, \mathrm{d}y} \tag{NSI}$$

Observe that (NSI) follows from summing $\alpha_t^2 / (\alpha_t^2 + \beta_t^2)$ times (TSI) and $\beta_t^2 / (\alpha_t^2 + \beta_t^2)$ times (CSI).

**Target Score Matching** The Target Score Matching loss was proposed by (Bortoli et al., 2024), and is a consequence of the target score identity (TSI), and the fact that when $p_{\text{data}}$ is the tilted distribution (9), its score is $\nabla \log p_{\text{data}} = \nabla \log p^{\text{base}} + \nabla r(x)$. The loss function reads

$$\mathcal{L}_{\text{TSM}}(\hat{s}) = \mathbb{E}_{\substack{Y \sim p_{\text{data}}, \\ \bar{X}_t = \alpha_t Y + \beta_t \varepsilon}} \left[ \int_0^1 \| \hat{s}(\bar{X}_t, t) - \tfrac{1}{\alpha_t} (\nabla \log p^{\text{base}}(Y) + \nabla r(Y)) \|^2 w(\bar{X}_t, t) \, \mathrm{d}t \right], \tag{19}$$

where $w : \mathbb{R}^d \times [0, 1] \to (0, +\infty)$ is an arbitrary weight function.

**Conditional Score Matching** The well-known Conditional Score Matching loss was used by the foundational works on diffusion models (Ho et al., 2020; Song & Ermon, 2019), and can be derived from the conditional score identity (CSI) analogously to the Target Score Matching loss. In our notation, the loss reads

$$\mathcal{L}_{\text{CSM}}(\hat{s}) = \mathbb{E}_{\substack{Y \sim p_{\text{data}}, \\ \bar{X}_t = \alpha_t Y + \beta_t \varepsilon}} \left[ \int_0^1 \| \hat{s}(\bar{X}_t, t) - \tfrac{\bar{X}_t + \alpha_t Y}{\beta_t^2} \|^2 w(\bar{X}_t, t) \, \mathrm{d}t \right], \tag{20}$$

where $w$ is an arbitrary weight function as in equation (91).

**Novel Score Matching** The novel score matching loss was introduced by Phillips et al. (2024), and can be derived from the novel score identity (NSI) analogously to the Target Score Matching loss. The loss function reads:

$$\mathcal{L}_{\text{NSM}}(\hat{s}) = \mathbb{E}_{\substack{Y \sim p_{\text{data}}, \\ \bar{X}_t = \alpha_t Y + \beta_t \varepsilon}} \left[ \int_0^1 \| \hat{s}(\bar{X}_t, t) - \tfrac{\alpha_t (\nabla \log p^{\text{base}}(Y) + \nabla r(Y)) - (\bar{X}_t - \alpha_t Y)}{\alpha_t^2 + \beta_t^2} \|^2 w(\bar{X}_t, t) \, \mathrm{d}t \right]. \tag{21}$$

**Iterated Denoising Energy Matching** A fourth loss function which assumes access to the density $p_{\text{data}}$ and thus can only be used for sampling is the iDEM loss (Akhound-Sadegh et al., 2024), defined as

$$\mathcal{L}_{\text{iDEM}}(\hat{s}) = \mathbb{E}_{\substack{Y \sim p_{\text{data}}, \\ \bar{X}_t = \alpha_t Y + \beta_t \varepsilon, \\ (\varepsilon_i)_{i=1}^n \sim \mathcal{N}(0, \mathrm{I})}} \left[ \int_0^1 \| \hat{s}(\bar{X}_t, t) - \tfrac{1}{\alpha_t} \tfrac{\sum_{i=1}^n \nabla \log p_{\text{data}}(\frac{\bar{X}_t - \beta_t \varepsilon_i}{\alpha_t}) p_{\text{data}}(\frac{\bar{X}_t - \beta_t \varepsilon_i}{\alpha_t})}{\sum_{i=1}^n p_{\text{data}}(\frac{\bar{X}_t - \beta_t \varepsilon_i}{\alpha_t})} \|^2 \, \mathrm{d}t \right]. \tag{22}$$

It can be viewed as a biased approximation of the quantity

$$\mathbb{E}_{Y \sim p_{\text{data}}, \bar{X}_t = \alpha_t Y + \beta_t \varepsilon} \left[ \int_0^1 \| \hat{s}(\bar{X}_t, t) - \mathbb{E} \left[ \tfrac{1}{\alpha_t} (\nabla \log p^{\text{base}}(Y) + \nabla r(Y)) \mid \bar{X}_t \right] \|^2 \, \mathrm{d}t \right], \tag{23}$$

which is equal to the bias term of the Target Score Matching loss (19) (see (25)).

**Solving the exponential tilting problem with score-based methods** Naturally, the score-based methods presented above can also be used to sample from the tilted distribution $p_{\text{data}}(x) \propto p^{\text{base}}(x) \exp(r(x))$ provided that we have samples from this distribution, which may be obtained e.g. using SMC methods (Phillips et al., 2024). In practice, the score-based methods that make explicit use of the score $\nabla \log p_{\text{data}} = \nabla \log p^{\text{base}}(x) + \nabla r(x)$ can be used even when we initially are only able to sample from $p^{\text{base}}$, as it is expected that if we keep on sampling using the learned model, the generated distribution will converge to the tilted distribution. In fact, some methods such as iDEM are introduced to work on-policy in this fashion. However, there are no theoretical guarantees that the on-policy versions of these algorithms converge to the tilted distribution.

### 3.3 ALGORITHM COMPARISON THROUGH BIAS-VARIANCE DECOMPOSITIONS

In this section, we show that all the loss functions presented in Sec. 3 can be written as the sum of a KL divergence term between a learned process and an optimal process (the *bias* term), and a positive term that has no contribution to the expected gradient (the *variance* term). To compare all algorithms on an equal footing, we write the learned fine-tuned generative SDE with memoryless schedule $\sigma(t) = \sqrt{2\eta_t}$, i.e. $dX_t = v_{\text{ft}}(X_t, t)\,dt + \sqrt{2\eta_t}\,dB_t$, with $X_0 \sim \mathcal{N}(0, \sigma_0^2 \mathrm{I})$. We set the importance weights of each loss function such that it takes the form:

$$\mathcal{L}(v_{\text{ft}}) = \mathbb{E}\big[\tfrac{1}{2}\int_0^1 \big\|v_{\text{ft}}(X_t, t) - \xi(t, X)\big\|^2 \tfrac{1}{2\eta_t}\,dt\big], \tag{24}$$

where the process $X$ and the vector field $\xi$ depend on the specific algorithm (see Prop. 3.2 and Prop. 3.3). Observe that this general loss can be further rewritten as

$$\mathcal{L}(v_{\text{ft}}) = \underbrace{\mathbb{E}\big[\int_0^1 \big\|v_{\text{ft}}(X_t, t) - \mathbb{E}[\xi(t, X)|X_t]\big\|^2 \tfrac{1}{2\eta_t}\,dt\big]}_{\text{Bias}} + \underbrace{\mathbb{E}\big[\int_0^1 \big\|\xi(t, X) - \mathbb{E}[\xi(t, X)|X_t]\big\|^2 \tfrac{1}{2\eta_t}\,dt\big]}_{\text{Variance}}. \tag{25}$$

In certain instances, the bias term as the (forward or reverse) KL divergence between the path measures of the optimal and learned processes, through the Girsanov theorem. The variance term does not contribute to the expected gradient, but it adds noise to the empirical gradient; it is desirable to minimize its contribution.

**Proposition 3.2** (Bias-variance decomposition for Adjoint Matching and Sampling). *The Adjoint Matching and Sampling loss functions in* (14) *and* (17) *fit the general form* (24) *by setting $X = X^{\bar{u}}, \bar{X}^{\bar{u}}$ resp., and identifying $v(x,t)/\sqrt{2\eta_t} = u(x,t)$, $\xi(t,X)/\sqrt{2\eta_t} = \sqrt{2\eta_t}\tilde{a}(t,X)$. Assume that the density $p^{\text{base}}$ is $\frac{1}{\sigma_1^2}$-strongly log-concave, i.e. $-\nabla^2 \log p^{\text{base}}(x) \succeq \frac{1}{\sigma_1^2}\mathrm{I}$. Then,*

(i) *The variance term in* (25) *admits the upper-bound $\frac{\sigma_1^2}{2}\mathbb{E}_{Y \sim p^{\text{base}}}[\|\nabla_x r(Y)\|^2]$ for three subcases of Flow Matching considered in Sec. 2.1.*

(ii) *If $p^{\text{base}} = \mathcal{N}(0, \sigma_1^2 \mathrm{I})$ as in AS, the variance term is $\frac{\sigma_1^2}{2}\mathrm{Tr}\big(\mathrm{Cov}_{Y \sim \mathcal{N}(0,\sigma_1^2 \mathrm{I})}[\nabla_x r(Y)]\big)$.*

(iii) *If we consider the AM loss function in which the expectation is with respect to the optimal process* (4)*, then the bias term is equal to the KL divergence $\mathrm{KL}(\mathbb{P}^\star || \mathbb{P}^u)$ between the optimal measure and the measure of $X^u$.*

As we remark in Sec. 3.1, the strong convexity assumption on $p^{\text{base}}$ is strong for the fine-tuning case, but a similar behavior is expected to hold in general. Note the AM loss function with optimal process expectation can be implemented via a Girsanov factor, and was first studied by (Domingo-Enrich et al., 2024, App. C.4) under the name SOCM-Adjoint method. Next, we prove a similar result for score matching methods.

**Proposition 3.3** (Bias-variance decomposition for score matching methods). *The Target, Conditional and Novel Score Matching loss functions in* (19)*,* (20) *and* (21) *fit the general form* (24) *by setting $X = \bar{X}$ (the reference flow) and weight function $w(x,t) = \eta_t$, identifying $\hat{s}(x,t) = \frac{v_{\text{ft}}(x,t) - \kappa_t x}{2\eta_t}$, and $\xi(t,X) = \kappa_t \bar{X}_t + \frac{2\eta_t}{\alpha_t}\big(\nabla \log p_{\text{base}}(Y) + \nabla r(Y)\big)$, $\xi(t,X) = \kappa_t \bar{X}_t - \frac{2\eta_t(\bar{X}_t - \alpha_t Y)}{\beta_t^2}$, and $\xi(t,X) = \kappa_t \bar{X}_t + \frac{2\eta_t\big(\alpha_t(\nabla \log p_{\text{base}}(Y) + \nabla r(Y)) - (\bar{X}_t - \alpha_t Y)\big)}{\alpha_t^2 + \beta_t^2}$, respectively. Moreover,*

(i) *For TSM and CSM, the variance term in* (25) *is infinite for the three subcases of Flow Matching considered in Sec. 2.1.*

(ii) *For NSM, the variance term admits the bounds shown in Tab. 1 (the bounds have a removable discontinuity at $\sigma_0 = 1$).*

(iii) *For TSM, CSM and NSM, the bias term is equal to the KL divergence $\mathrm{KL}(\mathbb{P}^\star || \mathbb{P}^{\hat{s}})$ between the optimal measure and the measure of $X^{\hat{s}}$, the process induced by $\hat{s}$.*

The reason that the variance term is infinite for TSM is that its integrand blows up at $t = 0$, and for CSM it blows up both at $t = 0$ and $t = 1$. Surprisingly, NSM manages to avoid all blow-ups. The infinite variance terms for TSM and CSM means that these methods cannot be run with weight

$w(x,t) = \eta_t$, i.e. by optimizing the expected loss function $\mathrm{KL}(\mathbb{P}^\star||\mathbb{P}^{\hat{s}})$, as the noise would be infinite (up to numerical aspects). Of course, that does not preclude using different weight functions, but those other weight functions will likely not yield a loss function with a probabilistic interpretation in terms of path measures like $w(x,t) = \eta_t$ does. Lastly, iDEM has a similar behavior to TSM, by the connection that we point out in (23).

| Method | Föllmer | DDIM/DDPM | Rectified Flow |
|---|---|---|---|
| AM/AS | $\frac{\sigma_1^2}{2}\mathbb{E}\big[\|\nabla_x r(Y)\|^2\big]$ | $\frac{\sigma_1^2}{2}\mathbb{E}\big[\|\nabla_x r(Y)\|^2\big]$ | $\frac{\sigma_1^2}{2}\mathbb{E}\big[\|\nabla_x r(Y)\|^2\big]$ |
| TSM | $+\infty$ | $+\infty$ | $+\infty$ |
| CSM | $+\infty$ | $+\infty$ | $+\infty$ |
| NSM | $\big(\frac{\sigma_0^2}{2(1-\sigma_0^2)} + \frac{\sigma_0^4}{2(1-\sigma_0^2)^2}\log(\sigma_0^2)\big)$ $\times\mathbb{E}\big[\|\nabla\log p^{\mathrm{base}}(Y)+\nabla r(Y)+Y\|^2\big]$ | $-\frac{\sigma_0^2}{2(1-\sigma_0^2)}\log(\sigma_0^2)$ $\times\mathbb{E}\big[\|\nabla\log p^{\mathrm{base}}(Y)+\nabla r(Y)+Y\|^2\big]$ | $\frac{\sigma_0^2(\pi-2)}{4}$ $\times\mathbb{E}\big[\|\nabla\log p^{\mathrm{base}}(Y)+\nabla r(Y)+Y\|^2\big]$ |
| iDEM | $+\infty$ | $+\infty$ | $+\infty$ |

Table 1: Comparison of the variance term bounds for each method.

## 4 THERMODYNAMICS-BASED ALGORITHMS AND ANALYSIS

In this section, we adapt the methods from (Vargas et al., 2024) and (Albergo & Vanden-Eijnden, 2025) to solve the thermodynamics formulation of the exponential tilting problem as described in Sec. 2.2. Relying on similar tools, we also prove novel versions of the escorted Crooks fluctuation theorem and the Jarzynski equality tailored to the exponential tilting dynamics.

For an arbitrary vector field $v$, let $X^v$ be the solution of the SDE (13), and let $\overleftarrow{X}^v$ be the solution of

$$\mathrm{d}\overleftarrow{X}_t^v = \big(\overleftarrow{b}_\sigma^r(\overleftarrow{X}_t^v, t) + v(\overleftarrow{X}_t^v, t)\big)\,\mathrm{d}t + \sigma(t)\overleftarrow{\mathrm{d}W_t}, \quad \begin{cases} \overleftarrow{X}_0^v \sim p_1^\star \propto p_1^{\mathrm{base}}\exp(r_1), \\ \overleftarrow{b}_\sigma^r(x,t) := \kappa_t x + \big(\eta_t - \frac{\sigma(t)^2}{2}\big)\big(\mathfrak{s}_t(x) + \nabla r_t(x)\big). \end{cases}$$

(26)

where $\overleftarrow{\mathrm{d}W_t}$ denotes the backward Itô differential, i.e. $\overleftarrow{X}_t^v$ is the continuous-time limit of the backward Euler-Maruyama update $y_{\ell-1} = y_\ell + \Delta t\gamma_{\ell\Delta t}^-(y_\ell) + \sqrt{\Delta t}\sigma(\ell\Delta t)\xi_\ell$, $\xi_\ell \sim \mathcal{N}(0,I)$. Let $\vec{\mathbb{P}}^v$ and $\overleftarrow{\mathbb{P}}^v$ be the the path measures of $X^v$ and $\overleftarrow{X}^v$. If $v = v^\star$ is such that $X^{v^\star} \sim p_t^\star(x) \propto p_t^{\mathrm{base}}(x)\exp(r_t(x))$, Nelson's relation (Prop. A.2) implies that $\vec{\mathbb{P}}^{v^\star} = \overleftarrow{\mathbb{P}}^{v^\star}$. By the reverse implication of Nelson's relation, the reciprocal statement also holds: if $\vec{\mathbb{P}}^v = \overleftarrow{\mathbb{P}}^v$, then $X^v \sim p_t^\star(x) \propto p_t^{\mathrm{base}}(x)\exp(r_t(x))$ and thus $v = v^\star$. Hence, any divergence $D$ on path measures gives rise to a loss function $\mathcal{L}_D(v) = D(\vec{\mathbb{P}}^v||\overleftarrow{\mathbb{P}}^v)$ whose only minimizer is $v = v^\star$. The following proposition shows the loss functions resulting from the KL divergence and the log-variance divergence. Its proof, which involves computing $\log\frac{\mathrm{d}\overleftarrow{\mathbb{P}}^v}{\mathrm{d}\vec{\mathbb{P}}^v}(X^v)$, can be found in (C.1).

**Proposition 4.1** (CMCD loss function for exponential tilting). *The CMCD loss functions for the exponential tilting problem based on the KL divergence the log-variance divergence read, respectively:*

$$\mathcal{L}_{\mathrm{KL-CMCD}}(v) = \mathbb{E}\big[\log\tfrac{\mathrm{d}\overleftarrow{\mathbb{P}}^v}{\mathrm{d}\vec{\mathbb{P}}^v}(X^v)\big]$$

$$= \mathbb{E}\Big[-\int_0^1 \sigma(t)^{-1}\langle v(X_t^v,t)+(\eta_t-\tfrac{\sigma(t)^2}{2})\nabla r_t(X_t^v), \overleftarrow{\mathrm{d}W_t}\rangle - r(X_1^v)$$

$$+ \int_0^1 \big(-\langle v(X_t^v,t), \mathfrak{s}_t(X_t^v)\rangle + \big(\tfrac{\sigma(t)^2}{2}-\eta_t\big)\langle\nabla r_t(X_t^v), \mathfrak{s}_t(X_t^v)\rangle + \tfrac{\sigma(t)^2}{2}\|\nabla r_t(X_t^v)\|^2\big)\,\mathrm{d}t\Big],$$

$$\mathcal{L}_{\mathrm{Var-CMCD}}(v) = \mathrm{Var}\Big[\log\tfrac{\mathrm{d}\overleftarrow{\mathbb{P}}^v}{\mathrm{d}\vec{\mathbb{P}}^v}(X^v)\Big] = \mathrm{Var}\Big[r_0(Y_0^v) - r_1(Y_1^v)$$

$$+ \int_0^1\Big[-\langle v(Y_t^v,t), \mathfrak{s}_t(Y_t^v)\rangle + \big(\tfrac{\sigma(t)^2}{2}-\eta_t\big)\langle\nabla r_t(Y_t^v), \mathfrak{s}_t(Y_t^v)\rangle + \tfrac{\sigma(t)^2}{2}\|\nabla r_t(Y_t^v)\|^2\Big]\,\mathrm{d}t$$

$$+ \int_0^1\Big[\sigma(t)^{-1}\big(\langle v(Y_t^v,t)+\eta_t\nabla r_t(Y_t^v), \overrightarrow{\mathrm{d}W_t}\rangle - \langle v(Y_t^v,t)+\eta_t\nabla r_t(Y_t^v), \overleftarrow{\mathrm{d}W_t}\rangle\big)$$

$$+ \tfrac{\sigma(t)}{2}\big(\langle\nabla r_t(Y_t^v), \overrightarrow{\mathrm{d}W_t}\rangle + \langle\nabla r_t(Y_t^v), \overleftarrow{\mathrm{d}W_t}\rangle\big)\Big]\Big].$$

(27)

The Crooks fluctuation theorem (Crooks, 1999) is a fundamental result in non-equilibrium thermodynamics that expresses the Radon-Nikodym derivative between a pair of forward and backward path measures in terms of a difference of free energies (or logarithm of normalizing constants) and a work functional. The following result, proven in App. C.2, provides an analogous expression for the path measures of $X^v$ and $\overleftarrow{X}^v$.

**Proposition 4.2** (Controlled Crooks fluctuation theorem for exponential tilting). *For an arbitrary process in the support of $\vec{\mathbb{P}}^v$ and/or $\overleftarrow{\mathbb{P}}^v$, the Radon-Nikodym derivative (RND) between $\vec{\mathbb{P}}^v$ and $\overleftarrow{\mathbb{P}}^v$ at reads*

$$\frac{\mathrm{d}\vec{\mathbb{P}}^v}{\mathrm{d}\overleftarrow{\mathbb{P}}^v}(Y) = \exp\Big( -\log \mathbb{E}_{\mathcal{N}(0,\beta_0^2 \mathrm{I})}[\exp(r_0)] + \log \mathbb{E}_{p^{\mathrm{base}}}[\exp(r_1)]$$
$$- \int_0^1 \Big( \langle \kappa_t Y_t + 2\eta_t \mathfrak{s}_t(Y_t), \nabla r_t(Y_t) \rangle + \langle v(Y_t,t), \mathfrak{s}_t(Y_t) + \nabla r_t(Y_t) \rangle + \partial_t r_t(Y_t)$$
$$+ \eta_t \|\nabla r_t(Y_t)\|^2 + \eta_t \,\Delta r_t(Y_t) + \nabla \cdot v(Y_t,t) \Big) \,\mathrm{d}t \Big). \tag{28}$$

Drawing analogy to the standard controlled Crooks fluctuation theorem (Vargas et al., 2024; Zhong et al., 2024), we can treat $-\log \mathbb{E}_{\mathcal{N}(0,\beta_0^2 \mathrm{I})}[\exp(r_0)] + \log \mathbb{E}_{p^{\mathrm{base}}}[\exp(r_1)]$ as a free energy difference and the remaining terms as a generalized work functional. Taking the expectation with respect to $Y \in \vec{\mathbb{P}}^v$ of the multiplicative inverse of both sides of (28) yields an analog of the escorted Jarzynski equality, first proposed by Vaikuntanathan & Jarzynski (2008).

**Proposition 4.3** (Escorted Jarzynski equality for exponential tilting). *The free energy difference admits the expression*

$$\log\Big( \frac{\mathbb{E}_{p^{\mathrm{base}}}[\exp(r_1)]}{\mathbb{E}_{\mathcal{N}(0,\beta_0^2 \mathrm{I})}[\exp(r_0)]} \Big) = \log \mathbb{E}_{\vec{\mathbb{P}}^v}\Big[ \exp\Big( \int_0^1 \Big( \langle \kappa_t Y_t + 2\eta_t \mathfrak{s}_t(Y_t), \nabla r_t(Y_t) \rangle + \langle v(Y_t,t), \mathfrak{s}_t(Y_t) +$$
$$\nabla r_t(Y_t) \rangle + \partial_t r_t(Y_t) + \eta_t \|\nabla r_t(Y_t)\|^2 + \eta_t \,\Delta r_t(Y_t) + \nabla \cdot v(Y_t,t) \Big) \,\mathrm{d}t \Big) \Big]. \tag{29}$$

Taking an expectation of the squared log-RND (28) and applying Jensen's inequality yields an analog of the NETS loss, first introduced by Albergo & Vanden-Eijnden (2025) in the standard thermodynamic setting. The proof is in App. C.3.

**Proposition 4.4** (NETS loss function for exponential tilting). *Given an arbitrary process $Y$, the PINN (physics informed neural network) NETS loss for exponential tilting reads*

$$\mathcal{L}_{\mathrm{NETS}}(v,F) = \mathbb{E}\Big[ \int_0^1 \Big( \langle \kappa_t Y_t + 2\eta_t \mathfrak{s}_t(Y_t), \nabla r_t(Y_t) \rangle + \langle v(Y_t,t), \mathfrak{s}_t(Y_t) + \nabla r_t(Y_t) \rangle$$
$$+ \partial_t r_t(Y_t) + \eta_t \|\nabla r_t(Y_t)\|^2 + \eta_t \,\Delta r_t(Y_t) + \nabla \cdot v(Y_t,t) - \partial_t F_t \Big)^2 \,\mathrm{d}t \Big], \tag{30}$$

*and it satisfies that $\mathcal{L}_{\mathrm{NETS}}(v,F) \le \mathbb{E}\big[ \big( \log \frac{\mathrm{d}\vec{\mathbb{P}}^v}{\mathrm{d}\overleftarrow{\mathbb{P}}^v}(Y) \big)^2 \big]$.*

## 5 Experiments

While the scope of our paper is general, as we cover fine-tuning and sampling, and many different algorithms, in this section we focus on the performance of Adjoint Matching for fine-tuning Stable Diffusion 1.5 and Stable Diffusion 3 with ImageReward (Xu et al., 2023) as the reward model. We fine-tune using the 10000 prompts considered by Xu et al. (2023) and report metrics computed on their 100-prompt validation dataset (generating 10 images per prompt).

In Fig. 1 we plot the trade-offs Astolfi et al. (2024) between DreamSim variance (Fu et al. (2023), a metric that measures per-prompt diversity) and ImageReward, CLIPScore (Hessel et al., 2021) and HPSv2 (Wu et al., 2023). Our results follow the same trend as those of Domingo-Enrich et al. (2025), which carried out similar experiments on a proprietary base model. We perform inference with $\eta = 0$ (no noise) and $\eta = 1$ (memoryless, $\sigma(t) = \sqrt{2\eta_t}$), and with two schedules: the default DDIM schedule and the schedule used during fine-tuning. Remarkably, $\eta = 0$ performs better w.r.t ImageReward and HPS, and $\eta = 1$ is better at CLIPScore.

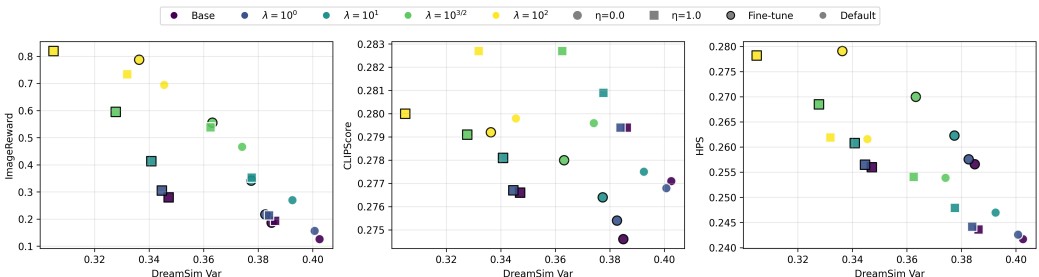

Figure 1: Quality metrics for Stable Diffusion 1.5 fine-tuned with Adjoint Matching.

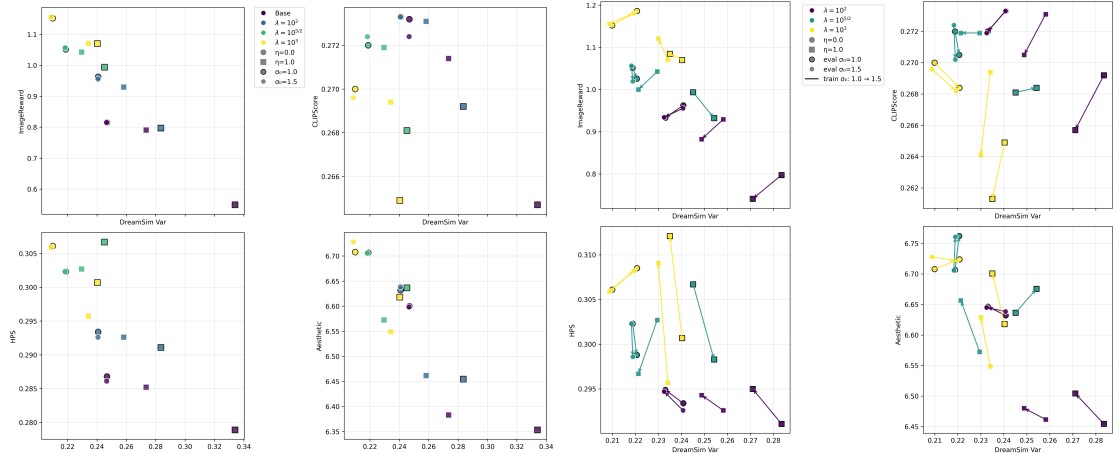

(a) Results for the base model and models fine-tuned at $\sigma_0^2 = 1$ and $\lambda \in \{10^2, 10^{5/2}, 10^3\}$, with inference at $\eta \in \{0, 1\}$ and $\sigma_0 = \{1, 1.5\}$.

(b) Results for models fine-tuned at $\sigma_0^2 \in \{1, 1.5\}$ and $\lambda \in \{10^2, 10^{5/2}, 10^3\}$, and different inference parameters. Points linked only differ in training $\sigma_0^2$.

Figure 2: Quality metrics for Stable Diffusion 3 fine-tuned with Adjoint Matching.

In Fig. 2(*left*) we plot the same trade-offs for Stable Diffusion 3, and include Aesthetic Score (LAION, 2024) as well. Arguably, $\eta = 1$ outperforms $\eta = 0$ in this case, as most points on the Pareto front use the former. In Fig. 2(*right*) we ablate the choice of the initial variance $\sigma_0^2$; as we show in App. D.1, we can simulate a generative SDE with a rescaled noise schedule $\sigma$ by reusing the pretrained vector field, which was we learned at $\sigma_0^2$. In the figure, points linked by an arrow correspond to settings which only differ by the training $\sigma_0^2$, the tail being for $\sigma_0^2 = 1$ and the head being for $\sigma_0^2 = 1.5$. We perform inference at both $\sigma_0^2 = 1$ and $1.5$. The results are inconclusive, which is consistent with Prop. 3.2, that states that the (bound on the) variance term for Adjoint Matching is independent of $\sigma_0$.

## 6 DISCUSSION

We introduced new developments that help us understand algorithms for fine-tuning and sampling with diffusion and flow models. We performed experiments to validate some of our findings. A direction of future work is to develop methods that leverage both the SOC and thermodynamics perspectives.

**Limitations**. We only include experiments on fine-tuning text-to-image diffusion models. We will leave experiments on other tasks such as protein design and molecule generation for future work. We also only include experiments on SOC and score matching-based approaches and leave experiments on the proposed thermodynamic-inspired approaches as future work. Our framework is not comprehensive as it does not include recent reward fine-tuning and sampling algorithms such as Liu et al. (2025b); Zhang et al. (2024); Akhound-Sadegh et al. (2025).

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

# Contents

## A   USEFUL THEORETICAL RESULTS

**Lemma A.1** (Conditional and target score identities, Bortoli et al. (2024)). *Let $p_t$ be the density of the marginal $\bar{X}_t$ of the reference flow defined in equation* (1). *For any $\alpha > 0$, define the map $T_\alpha$ as $x \mapsto T_\alpha(x) = \alpha x$, and let $(T_\alpha)_{\#}p$ be the pushforward of the distribution $p$ by $T_\alpha$, whose density is $(T_{\alpha_t})_{\#}p_{\mathrm{data}}(x) = p_{\mathrm{data}}(x/\alpha_t)\frac{1}{\alpha_t}$. We write the density of the Gaussian $\mathcal{N}(\alpha_t y, \beta_t^2 I)$ as $\mathcal{N}(x; \alpha_t y, \beta_t^2 I) = \exp\left(-\frac{\|x - \alpha_t y\|^2}{2\beta_t^2}\right)/(2\pi\beta_t^2)^{d/2}$. Then,*

$$p_t(x) = \int_{\mathbb{R}^d} \mathcal{N}(x; \alpha_t y, \beta_t^2 I) p_{\mathrm{data}}(y)\, \mathrm{d}y = [(T_{\alpha_t})_{\#}p_{\mathrm{data}} * \mathcal{N}(0, \beta_t^2 I)](x), \tag{31}$$

*and we obtain the following identities:*

Conditional score identity:     $\nabla \log p_t(x) = -\frac{\int_{\mathbb{R}^d}(x - \alpha_t y)\mathcal{N}(x; \alpha_t y, \beta_t^2 I)\, p_{\mathrm{data}}(y)\, \mathrm{d}y}{\beta_t^2 \int_{\mathbb{R}^d} \mathcal{N}(x; \alpha_t y, \beta_t^2 I)\, p_{\mathrm{data}}(y)\, \mathrm{d}y},$     (CSI)

Target score identity:     $\nabla \log p_t(x) = \frac{\int_{\mathbb{R}^d} \nabla \log p_{\mathrm{data}}(y)\mathcal{N}(x; \alpha_t y, \beta_t^2 I)p_{\mathrm{data}}(y)\, \mathrm{d}y}{\alpha_t \int_{\mathbb{R}^d} \mathcal{N}(x; \alpha_t y, \beta_t^2 I)p_{\mathrm{data}}(y)\, \mathrm{d}y},$     (TSI)

Novel score identity:     $\nabla \log p_t(x) = \frac{\int_{\mathbb{R}^d}(\alpha_t \nabla \log p_{\mathrm{data}}(y) - (x - \alpha_t y))\mathcal{N}(x; \alpha_t y, \beta_t^2 I)\, p_{\mathrm{data}}(y)\, \mathrm{d}y}{(\alpha_t^2 + \beta_t^2) \int_{\mathbb{R}^d} \mathcal{N}(x; \alpha_t y, \beta_t^2 I)\, p_{\mathrm{data}}(y)\, \mathrm{d}y}.$

(NSI)

*Proof.* While this result is not new (see Bortoli et al. (2024) and references within), we provide a proof here because of its relevance. Observe that $Z_t = X_t - \alpha_t Y = \beta_t \varepsilon \sim \mathcal{N}(0, \beta_t^2 I)$. The following formula holds for the density of the noised fine-tuned distribution:

$$p_t(x) = \int_{\mathbb{R}^d} \frac{\exp\left(-\frac{\|x - \alpha_t y\|^2}{2\beta_t^2}\right)}{(2\pi\beta_t^2)^{d/2}} p_{\mathrm{data}}(y)\, \mathrm{d}y = \int_{\mathbb{R}^d} \frac{\exp\left(-\frac{\|x - \tilde{y}\|^2}{2\beta_t^2}\right)}{(2\pi\beta_t^2)^{d/2}} p_{\mathrm{data}}(\tilde{y}/\alpha_t)\frac{1}{\alpha_t}\, \mathrm{d}\tilde{y}$$

$$= \int_{\mathbb{R}^d} \frac{\exp\left(-\frac{\|x - \tilde{y}\|^2}{2\beta_t^2}\right)}{(2\pi\beta_t^2)^{d/2}} (T_\alpha)_{\#}p_{\mathrm{data}}(\tilde{y})\, \mathrm{d}\tilde{y} = [(T_{\alpha_t})_{\#}p_{\mathrm{data}} * \mathcal{N}(0, \beta_t^2 I)](x), \tag{32}$$

where we applied the change of variables $\tilde{y} = T_{\alpha_t}y = \alpha_t y$, and that the density of the pushforward $(T_{\alpha_t})_{\#}p_{\mathrm{data}}(x)$ is $p_{\mathrm{data}}(x/\alpha_t)\frac{1}{\alpha_t}$.

Equation (CSI) follows simply from $\nabla \log p_t(x) = \frac{\nabla p_t(x)}{p_t(x)}$ and from taking the gradient with respect to $x$ under the integration sign. We prove (TSI) next. If we let $x - \alpha_t y = z$, we have that $y = \frac{x - z}{\alpha_t}$, which implies that $|\frac{\mathrm{d}y}{\mathrm{d}z}| = \frac{1}{\alpha_t}$. Thus, we can write

$$p_t(x) = \int_{\mathbb{R}^d} \frac{\exp\left(-\frac{\|z\|^2}{2\beta_t^2}\right)}{(2\pi\beta_t^2)^{d/2}} p_{\mathrm{data}}\left(\frac{x - z}{\alpha_t}\right)\frac{1}{\alpha_t}\, \mathrm{d}z. \tag{33}$$

And

$$\nabla p_t(x) = \int_{\mathbb{R}^d} \frac{\exp\left(-\frac{\|z\|^2}{2\beta_t^2}\right)}{(2\pi\beta_t^2)^{d/2}} \nabla_x \big(p_{\text{data}}(\tfrac{x-z}{\alpha_t})\big) \tfrac{1}{\alpha_t} \, dz \tag{34}$$

$$= \int_{\mathbb{R}^d} \frac{\exp\left(-\frac{\|z\|^2}{2\beta_t^2}\right)}{(2\pi\beta_t^2)^{d/2}} \nabla_x \log p_{\text{data}}(\tfrac{x-z}{\alpha_t}) p_{\text{data}}(\tfrac{x-z}{\alpha_t}) \tfrac{1}{\alpha_t} \, dz \tag{35}$$

$$= \int_{\mathbb{R}^d} \frac{\exp\left(-\frac{\|z\|^2}{2\beta_t^2}\right)}{(2\pi\beta_t^2)^{d/2}} \nabla \log p_{\text{data}}(\tfrac{x-z}{\alpha_t}) p_{\text{data}}(\tfrac{x-z}{\alpha_t}) \tfrac{1}{\alpha_t^2} \, dz \tag{36}$$

$$= \int_{\mathbb{R}^d} \frac{\exp\left(-\frac{\|x-\alpha_t y\|^2}{2\beta_t^2}\right)}{(2\pi\beta_t^2)^{d/2}} \nabla \log p_{\text{data}}(y) p_{\text{data}}(y) \tfrac{1}{\alpha_t} \, dy. \tag{37}$$

Using that $\nabla \log p_t(x) = \frac{\nabla p_t(x)}{p_t(x)}$ concludes the proof of (TSI). To prove (NSI), we sum $\frac{\alpha_t^2}{\alpha_t^2+\beta_t^2}$ times the (TSI) identity and $\frac{\beta_t^2}{\alpha_t^2+\beta_t^2}$ times the (CSI) identity, and obtain:

$$\nabla \log p_t(x) = \frac{\int_{\mathbb{R}^d} \frac{1}{\alpha_t^2+\beta_t^2} \big(\alpha_t \nabla \log p_{\text{data}}(y) - (x-\alpha_t y)\big) \mathcal{N}(x; \alpha_t y, \beta_t^2 I) \, p_{\text{data}}(y) \, dy}{\int_{\mathbb{R}^d} \mathcal{N}(x; \alpha_t y, \beta_t^2 I) \, p_{\text{data}}(y) \, dy}. \tag{38}$$

$\square$

**Theorem A.2** (Convolution with a Gaussian Preserves Strong Log-Concavity). *Let $f : \mathbb{R}^d \to (0, \infty)$ be a density of the form $f(x) = e^{-\varphi(x)}$, where $\varphi \in C^2(\mathbb{R}^d)$ satisfies $\nabla^2\varphi(x) \succeq \gamma I$ for all $x \in \mathbb{R}^d$, i.e. $f$ is $\gamma$-strongly log-concave. Let $g(x) = (2\pi\sigma^2)^{-d/2} \exp\left(-\frac{\|x\|^2}{2\sigma^2}\right)$ be the density of $\mathcal{N}(0, \sigma^2 I)$. Define*
$$h = f * g.$$
*Then $h$ is $\frac{\gamma}{1+\gamma\sigma^2}$-strongly log-concave, i.e.*

$$\nabla^2\big[-\log h(x)\big] \succeq \tfrac{\gamma}{1+\gamma\sigma^2} I \quad \forall x \in \mathbb{R}^d. \tag{39}$$

*Proof.* Set $\psi(z) = \frac{\|z\|^2}{2\sigma^2}$, $F(y; x) = \varphi(y) + \psi(x - y)$. Then $h(x) = \int_{\mathbb{R}^d} e^{-F(y;x)} \, dy$. A standard "log-sum-exp" Hessian identity yields

$$\nabla^2\big[-\log h(x)\big] = \mathbb{E}\big[\nabla_x^2 F(Y; x)\big] - \text{Var}\big[\nabla_x F(Y; x)\big], \tag{40}$$

where $Y$ is drawn from the density proportional to $e^{-F(y;x)}$. We handle the two terms separately.

*1. Second-derivative term.*
$$\nabla_x^2 F(y; x) = \nabla^2\psi(x - y) = \tfrac{1}{\sigma^2} I, \tag{41}$$

so $\mathbb{E}[\nabla_x^2 F] = \tfrac{1}{\sigma^2} I$.

*2. Variance term.*
$$\nabla_x F(y; x) = \nabla\psi(x - y) = \tfrac{x-y}{\sigma^2}, \tag{42}$$

hence $\text{Var}[\nabla_x F] = \tfrac{1}{\sigma^4} \text{Cov}(Y)$.

Since $F(\cdot; x)$ is $(\gamma + 1/\sigma^2)$-strongly convex in $y$, the Brascamp–Lieb inequality gives

$$\text{Cov}(Y) \preceq \tfrac{1}{\gamma+1/\sigma^2} I = \tfrac{\sigma^2}{1+\gamma\sigma^2} I. \tag{43}$$

Therefore,

$$\text{Var}[\nabla_x F] \preceq \tfrac{1}{\sigma^4} \cdot \tfrac{\sigma^2}{1+\gamma\sigma^2} I = \tfrac{1}{\sigma^2(1+\gamma\sigma^2)} I. \tag{44}$$

Combining,

$$\nabla^2[-\log h(x)] \succeq \tfrac{1}{\sigma^2} I - \tfrac{1}{\sigma^2(1+\gamma\sigma^2)} I = \tfrac{\gamma}{1+\gamma\sigma^2} I. \tag{45}$$

Thus $h$ is $\frac{\gamma}{1+\gamma\sigma^2}$-strongly log-concave. $\square$

**Corollary A.3.** *Let $p_t$ be the density of the marginal $\bar{X}_t$ of the reference flow defined in equation* (1). *Suppose that the density $p_{\text{data}}$ is in $C^2(\mathbb{R}^d)$ and is $\gamma$-strongly log-concave, i.e. $-\nabla^2 \log p_{\text{data}}(x) \succeq \gamma I$ for all $x \in \mathbb{R}^d$. Then, for all $t \in [0, 1]$, the density $p_t$ is also in $C^2(\mathbb{R}^d)$ and $\frac{\gamma}{\alpha_t^2 + \gamma \beta_t^2}$-strongly log-concave.*

*Proof.* By equation (31) from Lemma A.1, we have that

$$p_t(x) = [(T_{\alpha_t})_\# p_{\text{data}} * \mathcal{N}(0, \beta_t^2 I)](x), \tag{46}$$

Observe that

$$\nabla \log[(T_{\alpha_t})_\# p_{\text{data}}(x)] = \frac{\nabla (T_{\alpha_t})_\# p_{\text{data}}(x)}{(T_{\alpha_t})_\# p_{\text{data}}(x)} = \frac{\nabla_x \left( p_{\text{data}}(x/\alpha_t) \frac{1}{\alpha_t} \right)}{p_{\text{data}}(x/\alpha_t) \frac{1}{\alpha_t}} = \frac{\nabla p_{\text{data}}(x/\alpha_t) \frac{1}{\alpha_t^2}}{p_{\text{data}}(x/\alpha_t) \frac{1}{\alpha_t}} = \frac{\nabla \log p_{\text{data}}(x/\alpha_t)}{\alpha_t},$$

$$\implies -\nabla^2 \log[(T_{\alpha_t})_\# p_{\text{data}}(x)] = -\frac{\nabla^2 \log p_{\text{data}}(x/\alpha_t)}{\alpha_t^2} \succeq \frac{\gamma}{\alpha_t^2} I, \tag{47}$$

where we used the $\gamma$-strong log-concavity of $p_{\text{data}}$. This shows that $(T_{\alpha_t})_\# p_{\text{data}}$ is $\frac{\gamma}{\alpha_t^2}$-strongly concave. Thus, a direct application of Theorem A.2 with $f = p_{\text{data}}$, $g = \mathcal{N}(0, \beta_t^2 I)$ implies that the strong log-concavity constant of $p_t$ is

$$\frac{\gamma}{1 + \gamma \sigma^2} = \frac{\frac{\gamma}{\alpha_t^2}}{1 + \frac{\gamma \beta_t^2}{\alpha_t^2}} = \frac{\gamma}{\alpha_t^2 + \gamma \beta_t^2}. \tag{48}$$

$\square$

**Theorem A.4** (Adaptation of Theorem 2.3 of Shaul et al. (2024)). *Let $(\alpha_t, \beta_t)$ and $(\tilde{\alpha}_r, \tilde{\beta}_r)$ be two pairs of flow matching coefficients, i.e., differentiable functions $\alpha_t, \tilde{\alpha}_r : [0, 1] \to [0, 1]$, and $\alpha_t, \tilde{\alpha}_r : [0, 1] \to [0, +\infty)$ satisfying:*

$$\alpha_0 = \tilde{\alpha}_0 = 0 = \beta_1 = \tilde{\beta}_1, \quad \alpha_1 = \tilde{\alpha}_1 = 1,$$
$$\text{and } \text{SNR}(t) := \frac{\alpha_t}{\beta_t}, \ \widetilde{\text{SNR}}(t) := \frac{\tilde{\alpha}_t}{\tilde{\beta}_t} \text{ are strictly increasing on } [0, 1). \tag{49}$$

*Define the scale-time transformation from $r$ to $t(r)$ via matching signal-to-noise ratios:*

$$\frac{\tilde{\alpha}_r}{\tilde{\beta}_r} = \frac{\alpha_{t(r)}}{\beta_{t(r)}}, \tag{50}$$

*and define the scale function*

$$s_r := \frac{\tilde{\beta}_r}{\beta_{t(r)}} = \frac{\tilde{\alpha}_r}{\alpha_{t(r)}}. \tag{51}$$

*Then, if $p_t$ is the marginal of $\bar{X}_t := \alpha_t Y + \beta_t \varepsilon$ and $\tilde{p}_t$ is the marginal of $\tilde{X}_t := \tilde{\alpha}_t Y + \tilde{\beta}_t \varepsilon$, for all $r \in [0, 1]$,*

$$\nabla \log \tilde{p}_r(x) = \frac{1}{s_r} \nabla \log p_{t(r)}(x/s_r). \tag{52}$$

*Proof.* First, we prove that $t(r)$ is well-defined. Observe that by assumption, SNR and $\widetilde{\text{SNR}}$ are bijective functions between $[0, 1) \to [0, +\infty)$, and that we can construct $t(r) := \text{SNR}^{-1}(\widetilde{\text{SNR}}(r))$.

Using the conditional score identity, we obtain that

$$\nabla \log p_t(x) = -\frac{\int_{\mathbb{R}^d} \frac{x - \alpha_t y}{\beta_t^2} \mathcal{N}(x; \alpha_t y, \beta_t^2 I) \, p_{\text{data}}(y) \, dy}{\int_{\mathbb{R}^d} \mathcal{N}(x; \alpha_t y, \beta_t^2 I) \, p_{\text{data}}(y) \, dy}, \qquad \nabla \log \tilde{p}_t(x) = -\frac{\int_{\mathbb{R}^d} \frac{x - \tilde{\alpha}_t y}{\tilde{\beta}_t^2} \mathcal{N}(x; \tilde{\alpha}_t y, \tilde{\beta}_t^2 I) \, p_{\text{data}}(y) \, dy}{\int_{\mathbb{R}^d} \mathcal{N}(x; \tilde{\alpha}_t y, \tilde{\beta}_t^2 I) \, p_{\text{data}}(y) \, dy}, \tag{53}$$

and observe that

$$\frac{\|x - \tilde{\alpha}_r y\|^2}{2 \tilde{\beta}_r^2} = \frac{\|x - s_r \alpha_{t(r)} y\|^2}{2 s_r^2 \beta_{t(r)}^2} = \frac{\|x/s_r - \alpha_{t(r)} y\|^2}{2 \beta_{t(r)}^2}, \quad \frac{x - \tilde{\alpha}_r y}{\tilde{\beta}_r^2} = \frac{x - s_r \alpha_{t(r)} y}{s_r^2 \beta_{t(r)}^2} = \frac{x/s_r - \alpha_{t(r)} y}{s_r \beta_{t(r)}^2} \tag{54}$$

which means that

$$\mathcal{N}(x; \tilde{\alpha}_r y, \tilde{\beta}_r^2 I) = \frac{\exp \left( -\frac{\|x - \tilde{\alpha}_r y\|^2}{2 \tilde{\beta}_r^2} \right)}{(2\pi \tilde{\beta}_r^2)^{d/2}} = \frac{\exp \left( -\frac{\|x/s_r - \tilde{\alpha}_{t(r)} y\|^2}{2 \beta_{t(r)}^2} \right)}{(2\pi s_r^2 \beta_{t(r)}^2)^{d/2}} = \frac{1}{s_r^d} \mathcal{N}(x/s_r; \tilde{\alpha}_{t(r)} y, \tilde{\beta}_{t(r)}^2 I), \tag{55}$$

and

$$\nabla \log \tilde{p}_r(x) = -\frac{\int_{\mathbb{R}^d} \frac{x - \tilde{\alpha}_r y}{\tilde{\beta}_r^2} \mathcal{N}(x; \tilde{\alpha}_r y, \tilde{\beta}_r^2 \mathrm{I}) \, p_{\text{data}}(y) \, \mathrm{d}y}{\int_{\mathbb{R}^d} \mathcal{N}(x; \tilde{\alpha}_r y, \tilde{\beta}_r^2 \mathrm{I}) \, p_{\text{data}}(y) \, \mathrm{d}y}$$

$$= -\frac{\int_{\mathbb{R}^d} \frac{x/s_r - \alpha_{t(r)} y}{s_r \beta_{t(r)}^2} \mathcal{N}(x/s_r; \tilde{\alpha}_{t(r)} y, \tilde{\beta}_{t(r)}^2 \mathrm{I}) \, p_{\text{data}}(y) \, \mathrm{d}y}{\int_{\mathbb{R}^d} \mathcal{N}(x/s_r; \tilde{\alpha}_{t(r)} y, \tilde{\beta}_{t(r)}^2 \mathrm{I}) \, p_{\text{data}}(y) \, \mathrm{d}y} = \frac{1}{s_r} \nabla \log p_{t(r)}(x/s_r). \tag{56}$$

$\square$

**Corollary A.5.** *Consider Rectified Flow with two values $\sigma_0$, $\tilde{\sigma}_0$. That is, we take the pairs $(\alpha_t, \beta_t)$ and $(\tilde{\alpha}_t, \tilde{\beta}_t)$ in Theorem A.4 such that:*

$$\alpha_t = \tilde{\alpha}_t = \bar{\alpha}_t, \qquad \beta_t = (1 - \bar{\alpha}_t)\sigma_0, \qquad \tilde{\beta}_t = (1 - \bar{\alpha}_t)\tilde{\sigma}_0. \tag{57}$$

*Let $\bar{\alpha}^{-1} : [0, 1] \to [0, 1]$ be the inverse of the function $\bar{\alpha}(t) := \bar{\alpha}_t$. Then, we have that*

$$t(r) = \bar{\alpha}^{-1}\left(\frac{\sigma_0 \bar{\alpha}_r}{\tilde{\sigma}_0(1 - \bar{\alpha}_r) + \sigma_0 \bar{\alpha}_r}\right), \qquad s_r = \frac{\tilde{\sigma}_0(1 - \bar{\alpha}_r) + \sigma_0 \bar{\alpha}_r}{\sigma_0}. \tag{58}$$

*and by Theorem A.4, if $p_t$ is the marginal of $\bar{X}_t := \alpha_t Y + \beta_t \varepsilon$ and $\tilde{p}_t$ is the marginal of $\tilde{X}_t := \tilde{\alpha}_t Y + \tilde{\beta}_t \varepsilon$,*

$$\nabla \log \tilde{p}_r(x) = \frac{1}{s_r} \nabla \log p_{t(r)}(x/s_r). \tag{59}$$

*Proof.* Observe that with these choices,

$$\text{SNR}(t) := \frac{\alpha_t}{\beta_t} = \frac{\bar{\alpha}_t}{(1 - \bar{\alpha}_t)\sigma_0}, \qquad \widetilde{\text{SNR}}(t) := \frac{\bar{\alpha}_t}{(1 - \bar{\alpha}_t)\tilde{\sigma}_0}. \tag{60}$$

We invert SNR:

$$y = \frac{\bar{\alpha}_t}{(1 - \bar{\alpha}_t)\sigma_0} \implies \sigma_0 y - \bar{\alpha}_t \sigma_0 y = \bar{\alpha}_t \implies \bar{\alpha}(t) := \bar{\alpha}_t = \frac{\sigma_0 y}{1 + \sigma_0 y} \implies \text{SNR}^{-1} = \bar{\alpha}^{-1}\left(\frac{\sigma_0 y}{1 + \sigma_0 y}\right). \tag{61}$$

Thus,

$$t(r) = \text{SNR}^{-1}\left(\widetilde{\text{SNR}}(r)\right) = \bar{\alpha}^{-1}\left(\frac{\sigma_0 \frac{\bar{\alpha}_r}{(1 - \bar{\alpha}_r)\tilde{\sigma}_0}}{1 + \sigma_0 \frac{\bar{\alpha}_r}{(1 - \bar{\alpha}_r)\tilde{\sigma}_0}}\right) = \bar{\alpha}^{-1}\left(\frac{\sigma_0 \bar{\alpha}_r}{\tilde{\sigma}_0(1 - \bar{\alpha}_r) + \sigma_0 \bar{\alpha}_r}\right), \tag{62}$$

and

$$s_r = \frac{\tilde{\alpha}_r}{\alpha_{t(r)}} = \frac{\bar{\alpha}_r}{\bar{\alpha}_{t(r)}} = \frac{\bar{\alpha}_r}{\frac{\sigma_0 \bar{\alpha}_r}{\tilde{\sigma}_0(1 - \bar{\alpha}_r) + \sigma_0 \bar{\alpha}_r}} = \frac{\tilde{\sigma}_0(1 - \bar{\alpha}_r) + \sigma_0 \bar{\alpha}_r}{\sigma_0}. \tag{63}$$

Applying Theorem A.4 yields the final result. $\square$

**Proposition A.1** (Forward–backward Radon–Nikodym derivatives, Prop. 2.2 of Vargas et al. (2024)). *Consider the SDEs*

$$\mathrm{d}Y_t = \gamma_t^+(Y_t) \, \mathrm{d}t + \sigma(t) \overrightarrow{\mathrm{d}W_t}, \qquad Y_0 \sim \Gamma_0 \implies (Y_t)_{0 \le t \le T} \sim \overrightarrow{\mathbb{P}}^{\Gamma_0, \gamma^+}, \tag{64}$$

$$\mathrm{d}Y_t = \gamma^-(Y_t) \, \mathrm{d}t + \sigma(t) \overleftarrow{\mathrm{d}W_t}, \qquad Y_T \sim \Gamma_T \implies (Y_t)_{0 \le t \le T} \sim \overleftarrow{\mathbb{P}}^{\Gamma_T, \gamma^-}, \tag{65}$$

$$\mathrm{d}Y_t = a_t(Y_t) \, \mathrm{d}t + \sigma(t) \overrightarrow{\mathrm{d}W_t}, \qquad Y_0 \sim \mu \implies (Y_t)_{0 \le t \le T} \sim \overrightarrow{\mathbb{P}}^{\mu, a}, \tag{66}$$

$$\mathrm{d}Y_t = b_t(Y_t) \, \mathrm{d}t + \sigma(t) \overleftarrow{\mathrm{d}W_t}, \qquad Y_T \sim \nu \implies (Y_t)_{0 \le t \le T} \sim \overleftarrow{\mathbb{P}}^{\nu, b}. \tag{67}$$

*Here, (64) and (66) are forward Itô SDEs, and (65) and (67) are backward Itô SDEs, i.e. (64) and (65) are the continuous-time limits of*

$$y_{\ell+1} = y_\ell + \Delta t \gamma_{\ell \Delta t}^+(y_\ell) + \sqrt{\Delta t} \sigma(\ell \Delta t) \xi_\ell, \qquad \xi_\ell \sim \mathcal{N}(0, I), \qquad y_0 \sim \Gamma_0,$$
$$y_{\ell-1} = y_\ell + \Delta t \gamma_{\ell \Delta t}^-(y_\ell) + \sqrt{\Delta t} \sigma(\ell \Delta t) \xi_\ell, \qquad \xi_\ell \sim \mathcal{N}(0, I), \qquad y_T \sim \Gamma_T. \tag{68}$$

*Suppose that*

$$\overrightarrow{\mathbb{P}}^{\Gamma_0, \gamma^+} = \overleftarrow{\mathbb{P}}^{\Gamma_T, \gamma^-}, \tag{69}$$

*and that it is absolutely continuous with respect to both $\overrightarrow{\mathbb{P}}^{\mu,a}$ and $\overleftarrow{\mathbb{P}}^{\nu,b}$. Then, $\overrightarrow{\mathbb{P}}^{\mu,a}$-almost surely, the corresponding Radon–Nikodym derivative can be expressed as*

$$\log \frac{\mathrm{d}\overrightarrow{\mathbb{P}}^{\mu,a}}{\mathrm{d}\overleftarrow{\mathbb{P}}^{\nu,b}}(Y) = \log \frac{\mathrm{d}\mu}{\mathrm{d}\Gamma_0}(Y_0) - \log \frac{\mathrm{d}\nu}{\mathrm{d}\Gamma_T}(Y_T) \tag{70}$$

$$+ \int_0^T \sigma(t)^{-2} \langle (a_t - \gamma_t^+)(Y_t), \overrightarrow{\mathrm{d}Y_t} - \tfrac{1}{2}(a_t + \gamma_t^+)(Y_t)\,\mathrm{d}t \rangle \tag{71}$$

$$- \int_0^T \sigma(t)^{-2} \langle (b_t - \gamma_t^-)(Y_t), \overleftarrow{\mathrm{d}Y_t} - \tfrac{1}{2}(b_t + \gamma_t^-)(Y_t)\,\mathrm{d}t \rangle. \tag{72}$$

**Proposition A.2** (Nelson's relation, Nelson (1967); Anderson (1982))**.** *Let $\overrightarrow{\mathbb{P}}^{\mu,a}$ and $\overleftarrow{P}^{\nu,b}$ be the path measures defined in Prop. A.1. For $\mu$ and $a$ of sufficient regularity, denote the time–marginals of the corresponding path measure by*

$$\overrightarrow{\mathbb{P}}^{\mu,a}_t =: \rho_t^{\mu,a}.$$

*Then we have*

$$\overrightarrow{\mathbb{P}}^{\mu,a} = \overleftarrow{\mathbb{P}}^{\nu,b} \quad \text{if and only if} \quad \nu = \overrightarrow{\mathbb{P}}^{\mu,a}_T \quad \text{and} \quad b_t = a_t - \sigma^2 \nabla \ln \rho_t^{\mu,a}, \quad \forall t \in (0, T].$$

# B PROOFS FOR THE SOC-BASED METHODS

## B.1 PROOF OF PROP. 3.1: BOUND ON THE NORM OF THE LEAN ADJOINT STATE

Given a matrix $M \in \mathbb{R}^{d \times d}$ and a point $x \in \mathbb{R}^d$, the Rayleigh quotient is defined as $R(M, x) = \frac{\langle x, Mx \rangle}{\langle x, x \rangle}$. The norm of the lean adjoint state $\tilde{a}(t, X^u)$ satisfies the following ODE:

$$\frac{\mathrm{d}}{\mathrm{d}t}\|\tilde{a}(t, X^u)\|^2 = 2\langle \tilde{a}(t, X^u), \tfrac{\mathrm{d}}{\mathrm{d}t}\tilde{a}(t, X^u) \rangle = -2\langle \tilde{a}(t, X^u), \nabla_x b(X_t, t)^\top \tilde{a}(t; X^{\bar{u}}) \rangle \tag{73}$$

$$= -2R(\nabla_x b(X_t, t)^\top, \tilde{a}(t, X^u))\|\tilde{a}(t; X^{\bar{u}})\|^2. \tag{74}$$

When we integrate this ODE backwards in time from 1 to $t \in [0, 1]$, we obtain that

$$\|\tilde{a}(t, X^u)\|^2 = \exp\left(2 \int_t^1 R(\nabla_x b(X_s, s)^\top, \tilde{a}(s, X^u))\,\mathrm{d}s\right) \|\nabla_x r(X_1^u)\|^2. \tag{75}$$

Since $\tilde{a}(t, X^u)$ appears in the the regression target vector field of the Adjoint Matching loss, it is desirable that the norm $\|\tilde{a}(t, X^u)\|$ is small. A way to obtain bounds on $\|\tilde{a}(t, X^u)\|$ is under the condition that $\mathrm{Sym}(\nabla_x b(X_t, t)) \preceq \chi_t I$ for some constant $\chi_t \in \mathbb{R}$, as in this case, since $R(\nabla_x b(X_t, t)^\top, \tilde{a}(t, X^u)) = R(\mathrm{Sym}(\nabla_x b(X_t, t)), \tilde{a}(t, X^u)) \leq \chi_t$, we get that

$$\|\tilde{a}(t, X^u)\|^2 \leq \exp\left(2 \int_t^1 \chi_s\,\mathrm{d}s\right) \|\nabla_x r(X_1^u)\|^2. \tag{76}$$

Observe that $\nabla b(x, t) = \kappa_t I + \left(\frac{\sigma(t)^2}{2} + \eta_t\right)\nabla \mathfrak{s}_t(x)$, and for the memoryless noise schedule $\sigma(t) = \sqrt{2\eta_t}$,

$$\nabla b(x, t) = \kappa_t I + 2\eta_t \nabla \mathfrak{s}_t(x). \tag{77}$$

Since $\eta_t > 0$, in order to obtain a bound of the form $\mathrm{Sym}(\nabla_x b(X_t, t)) \preceq \chi_t I$, we need a similar bound on $\mathrm{Sym}(\nabla_x \mathfrak{s}_t(X_t))$. Next, we show that such bounds are easy to obtain in the case in which the data distribution $p_{\mathrm{data}}$ is Gaussian or strongly log-concave. The former case, under which we can obtain an analytic expression of the score, is particularly relevant because it is the setting considered in Adjoint Sampling (Havens et al., 2025).

**The gradient $\nabla b$ for Gaussian data distributions** Now, if assume that $p_{\mathrm{data}} = \mathcal{N}(0, \sigma_1^2 I)$, as is the case in the sampling setting, we can compute $\mathfrak{s}_t$ explicitly through equation (31) of Lemma A.1. By equation (31), we obtain that

$$p_t(x) = \int_{\mathbb{R}^d} \frac{\exp\left(-\frac{\|x - \alpha_t y\|^2}{2\beta_t^2}\right)}{(2\pi\beta_t^2)^{d/2}} \frac{\exp\left(-\frac{\|y\|^2}{2\sigma^2}\right)}{(2\pi\sigma^2)^{d/2}}\,\mathrm{d}y = \frac{\exp\left(-\frac{\|x\|^2}{2(\beta_t^2 + \alpha_t^2 \sigma_1^2)}\right)}{\left(2\pi(\beta_t^2 + \alpha_t^2 \sigma_1^2)\right)^{d/2}}, \tag{78}$$

which implies $\mathfrak{s}_t(x) = \nabla \log p_t(x) = -\frac{x}{\beta_t^2 + \alpha_t^2 \sigma_1^2}$, and this means that

$$\nabla b(x, t) = \kappa_t I + 2\eta_t \nabla \mathfrak{s}_t(x) = \left(\kappa_t - \frac{2\eta_t}{\beta_t^2 + \alpha_t^2 \sigma_1^2}\right)I = \left(\frac{\dot{\alpha}_t}{\alpha_t} - \frac{2\beta_t\left(\frac{\dot{\alpha}_t}{\alpha_t}\beta_t - \dot{\beta}_t\right)}{\beta_t^2 + \alpha_t^2 \sigma_1^2}\right)I =: \chi_t I, \tag{79}$$

(i) Föllmer process:

$$\nabla b(x,t) = \big(\kappa_t - \tfrac{2\eta_t}{\beta_t^2+\alpha_t^2\sigma_1^2}\big)\mathrm{I} = \big(\tfrac{\dot{\bar{\alpha}}_t}{\bar{\alpha}_t} - \tfrac{\dot{\bar{\alpha}}_t\sigma_0^2}{\bar{\alpha}_t(1-\bar{\alpha}_t)\sigma_0^2+\bar{\alpha}_t^2\sigma_1^2}\big)\mathrm{I} = \tfrac{\dot{\bar{\alpha}}_t}{\bar{\alpha}_t}\big(1 - \tfrac{1}{1-\bar{\alpha}_t+\bar{\alpha}_t\sigma_1^2/\sigma_0^2}\big)\mathrm{I} =: \chi_t\mathrm{I},$$

$$\implies \int_t^1 \chi_s\,\mathrm{d}s = \int_t^1 \tfrac{\dot{\bar{\alpha}}_s}{\bar{\alpha}_s}\big(1 - \tfrac{1}{1-\bar{\alpha}_s+\bar{\alpha}_s\sigma_1^2/\sigma_0^2}\big)\,\mathrm{d}s = \log\big(\tfrac{\sigma_1^2}{(1-\bar{\alpha}_s)\sigma_0^2+\bar{\alpha}_s\sigma_1^2}\big),$$

$$\implies \|\tilde{a}(t,X^u)\| = \tfrac{\sigma_1^2}{(1-\bar{\alpha}_s)\sigma_0^2+\bar{\alpha}_s\sigma_1^2}\|\nabla_x r(X_1^u)\|. \tag{80}$$

If we set $\sigma_0^2 = \sigma_1^2$, we obtain $\nabla b(x,t) = 0$ and $\int_t^1 \chi_s\,\mathrm{d}s = 0$.

(ii) DDIM/DDPM:

$$\nabla b(x,t) = \big(\kappa_t - \tfrac{2\eta_t}{\beta_t^2+\alpha_t^2\sigma_1^2}\big)\mathrm{I} = \big(\tfrac{\dot{\bar{\alpha}}_t}{2\bar{\alpha}_t} - \tfrac{\frac{\dot{\bar{\alpha}}_t\sigma_0^2}{\bar{\alpha}_t}}{(1-\bar{\alpha}_t)\sigma_0^2+\bar{\alpha}_t\sigma_1^2}\big)\mathrm{I} = \tfrac{\dot{\bar{\alpha}}_t}{2\bar{\alpha}_t}\big(1 - \tfrac{2}{1-\bar{\alpha}_t+\bar{\alpha}_t\sigma_1^2/\sigma_0^2}\big)\mathrm{I} := \chi_t\mathrm{I},$$

$$\implies \int_t^1 \chi_s\,\mathrm{d}s = \int_t^1 \tfrac{\dot{\bar{\alpha}}_s}{2\bar{\alpha}_s}\big(1 - \tfrac{2}{1-\bar{\alpha}_s+\bar{\alpha}_s\sigma_1^2/\sigma_0^2}\big)\,\mathrm{d}s = \log\big(\tfrac{\sigma_1^2\sqrt{\bar{\alpha}_s}}{(1-\bar{\alpha}_s)\sigma_0^2+\bar{\alpha}_s\sigma_1^2}\big),$$

$$\implies \|\tilde{a}(t,X^u)\| = \tfrac{\sigma_1^2\sqrt{\bar{\alpha}_s}}{(1-\bar{\alpha}_s)\sigma_0^2+\bar{\alpha}_s\sigma_1^2}\|\nabla_x r(X_1^u)\|. \tag{81}$$

If we set $\sigma_0^2 = \sigma_1^2$, we obtain $\nabla b(x,t) = -\tfrac{\dot{\bar{\alpha}}_t}{2\bar{\alpha}_t}\mathrm{I}$, and $\int_t^1 \chi_s\,\mathrm{d}s = \tfrac{1}{2}\log(\bar{\alpha}_t)$, and $\|\tilde{a}(t,X^u)\| = \sqrt{\bar{\alpha}_s}\|\nabla_x r(X_1^u)\|$.

(iii) Rectified Flow:

$$\nabla b(x,t) = \big(\kappa_t - \tfrac{2\eta_t}{\beta_t^2+\alpha_t^2\sigma_1^2}\big)\mathrm{I} = \big(\tfrac{\dot{\bar{\alpha}}_t}{\bar{\alpha}_t} - \tfrac{2\frac{(1-\bar{\alpha}_t)\dot{\bar{\alpha}}_t\sigma_0^2}{\bar{\alpha}_t}}{(1-\bar{\alpha}_t)^2\sigma_0^2+\bar{\alpha}_t^2\sigma_1^2}\big)\mathrm{I} = \tfrac{\dot{\bar{\alpha}}_t}{\bar{\alpha}_t}\big(1 - \tfrac{2(1-\bar{\alpha}_t)}{(1-\bar{\alpha}_t)^2+\bar{\alpha}_t^2\sigma_1^2/\sigma_0^2}\big)\mathrm{I}, \tag{82}$$

$$\implies \int_t^1 \chi_s\,\mathrm{d}s = \int_t^1 \tfrac{\dot{\bar{\alpha}}_s}{\bar{\alpha}_s}\big(1 - \tfrac{2(1-\bar{\alpha}_s)}{(1-\bar{\alpha}_s)^2+\bar{\alpha}_s^2\sigma_1^2/\sigma_0^2}\big)\,\mathrm{d}s = \log\big(\tfrac{\sigma_1^2\bar{\alpha}_s}{(1-\bar{\alpha}_s)^2\sigma_0^2+\bar{\alpha}_s^2\sigma_1^2}\big), \tag{83}$$

$$\implies \|\tilde{a}(t,X^u)\| = \tfrac{\sigma_1^2\bar{\alpha}_s}{(1-\bar{\alpha}_s)^2\sigma_0^2+\bar{\alpha}_s^2\sigma_1^2}\|\nabla_x r(X_1^u)\|. \tag{84}$$

The maximizer of $\alpha \mapsto \log\big(\tfrac{\sigma_1^2\bar{\alpha}_s}{(1-\bar{\alpha}_s)^2\sigma_0^2+\bar{\alpha}_s^2\sigma_1^2}\big)$ is $\alpha^\star = \tfrac{\sigma_0}{\sqrt{\sigma_0^2+\sigma_1^2}}$, and the maximum value is $\tfrac{1+\sqrt{1+\sigma_1^2/\sigma_0^2}}{2}$. If we set $\sigma_0^2 = \sigma_1^2$, we obtain

$$\nabla b(x,t) = \tfrac{\dot{\bar{\alpha}}_t}{\bar{\alpha}_t}\big(1 - \tfrac{2(1-\bar{\alpha}_t)}{(1-\bar{\alpha}_t)^2+\bar{\alpha}_t^2}\big)\mathrm{I} = \tfrac{\dot{\bar{\alpha}}_t}{\bar{\alpha}_t}\tfrac{1-2\bar{\alpha}_t+2\bar{\alpha}_t^2-2(1-\bar{\alpha}_t)}{1-2\bar{\alpha}_t+2\bar{\alpha}_t^2}\mathrm{I} = -\tfrac{(1-2\bar{\alpha}_t)\dot{\bar{\alpha}}_t}{(1-2\bar{\alpha}_t+2\bar{\alpha}_t^2)\bar{\alpha}_t}\mathrm{I}, \tag{85}$$

$$\implies \int_t^1 \chi_s\,\mathrm{d}s = \log\big(\tfrac{\bar{\alpha}_s}{(1-\bar{\alpha}_s)^2+\bar{\alpha}_s^2}\big) \implies \|\tilde{a}(t,X^u)\| = \tfrac{\bar{\alpha}_s}{(1-\bar{\alpha}_s)^2+\bar{\alpha}_s^2}\|\nabla_x r(X_1^u)\|. \tag{86}$$

**The gradient $\nabla b$ for $\tfrac{1}{\sigma_1^2}$-strongly convex data distributions** Corollary A.3 proves that when $p_{\text{data}}$ is $\tfrac{1}{\sigma_1^2}$-strongly log-concave, then $p_t$ is also in $C^2(\mathbb{R}^d)$ and $\tfrac{1}{\beta_t^2+\alpha_t^2\sigma_1^2}$-strongly log-concave. Equivalently, for all $x \in \mathbb{R}^d$, $t \in [0,1]$,

$$-\nabla\mathfrak{s}_t(x) = -\nabla^2 \log p_t(x) \succeq \tfrac{\mathrm{I}}{\beta_t^2+\alpha_t^2\sigma_1^2}, \tag{87}$$

which means that

$$\nabla b(x,t) = \kappa_t\mathrm{I} + 2\eta_t\nabla\mathfrak{s}_t(x) \preceq \big(\kappa_t - \tfrac{2\eta_t}{\beta_t^2+\alpha_t^2\sigma_1^2}\big)\mathrm{I} = \big(\tfrac{\dot{\alpha}_t}{\alpha_t} - \tfrac{2\beta_t\left(\frac{\dot{\alpha}_t}{\alpha_t}\beta_t-\dot{\beta}_t\right)}{\beta_t^2+\alpha_t^2\sigma_1^2}\big)\mathrm{I}. \tag{88}$$

Observe that this upper-bound matches the right-hand side of (79). Hence, we obtain immediately that for the three subcases, all the equalities involving $\nabla b(x,t)$ and $\int_t^1 \chi_s\,\mathrm{d}s$ become inequalities.

## B.2 DERIVATION OF THE TARGET, CONDITIONAL AND NOVEL SCORE MATCHING LOSS FUNCTIONS

By the target score identity from Lemma A.1, the density $p_t$ of the marginal $\bar{X}_t$ of the reference flow in equation (1) satisfies:

$$\nabla \log p_t(x) = \tfrac{1}{\alpha_t}\tfrac{\int_{\mathbb{R}^d} \mathcal{N}(x;\alpha_t y,\beta_t^2\mathrm{I})\nabla \log p_{\text{data}}(Y)p_{\text{data}}(y)\,\mathrm{d}y}{\int_{\mathbb{R}^d} \mathcal{N}(x;\alpha_t y,\beta_t^2\mathrm{I})p_{\text{data}}(y)\,\mathrm{d}y} = \tfrac{1}{\alpha_t}\mathbb{E}[\nabla \log p_{\text{data}}(Y)\,|\,\bar{X}_t = x] \tag{89}$$

We use a well-known argument: for any $\mathbb{R}^d$-valued neural network $\hat{s}$,

$$\mathbb{E}_{Y\sim p_{\text{data}},\bar{X}_t=\alpha_t Y+\beta_t\varepsilon}\Big[\int_0^1\|\hat{s}(\bar{X}_t,t)-\nabla\log p_t(\bar{X}_t)\|^2\,\mathrm{d}t\Big]$$

$$=\mathbb{E}_{Y\sim p_{\text{data}},\bar{X}_t=\alpha_t Y+\beta_t\varepsilon}\Big[\int_0^1\|\hat{s}(\bar{X}_t,t)\|^2\,\mathrm{d}t-2\int_0^1\langle\hat{s}(\bar{X}_t,t),\nabla\log p_t(\bar{X}_t)\rangle\,\mathrm{d}t\Big]+\text{const.}$$

$$=\mathbb{E}_{Y\sim p_{\text{data}},\bar{X}_t=\alpha_t Y+\beta_t\varepsilon}\Big[\int_0^1\|\hat{s}(\bar{X}_t,t)\|^2\,\mathrm{d}t-\tfrac{2}{\alpha_t}\int_0^1\langle\hat{s}(\bar{X}_t,t),\mathbb{E}[\nabla\log p_{\text{data}}(Y)\,|\,\bar{X}_t]\rangle\,\mathrm{d}t\Big]+\text{const.}$$

$$=\mathbb{E}_{Y\sim p_{\text{data}},\bar{X}_t=\alpha_t Y+\beta_t\varepsilon}\Big[\int_0^1\|\hat{s}(\bar{X}_t,t)\|^2\,\mathrm{d}t-\tfrac{2}{\alpha_t}\int_0^1\langle\hat{s}(\bar{X}_t,t),\nabla\log p^{\text{base}}(Y)+\nabla r(Y)\rangle\,\mathrm{d}t\Big]+\text{const.}$$

$$=\mathbb{E}_{Y\sim p_{\text{data}},\bar{X}_t=\alpha_t Y+\beta_t\varepsilon}\Big[\int_0^1\|\hat{s}(\bar{X}_t,t)-\tfrac{1}{\alpha_t}\big(\nabla\log p^{\text{base}}(Y)+\nabla r(Y)\big)\|^2\,\mathrm{d}t\Big]+\text{const.}$$

$$(90)$$

where the third equality holds by (89). Hence,

$$\mathcal{L}_{\text{TSM}}(\hat{s})=\mathbb{E}[\mathcal{L}_{\text{TSM}}(\hat{s},\bar{X})]=\mathbb{E}_{Y\sim p_{\text{data}},\bar{X}_t=\alpha_t Y+\beta_t\varepsilon}\Big[\int_0^1\|\hat{s}(\bar{X}_t,t)-\tfrac{1}{\alpha_t}\big(\nabla\log p^{\text{base}}(Y)+\nabla r(Y)\big)\|^2\,\mathrm{d}t\Big].$$

$$(91)$$

Analogously to (89), the conditional score identity (CSI) and the novel score identity (NSI) yield

$$\nabla\log p_t(x)=-\frac{\int_{\mathbb{R}^d}(x-\alpha_t y)\mathcal{N}(x;\alpha_t y,\beta_t^2 I)\,p_{\text{data}}(y)\,\mathrm{d}y}{\beta_t^2\int_{\mathbb{R}^d}\mathcal{N}(x;\alpha_t y,\beta_t^2 I)\,p_{\text{data}}(y)\,\mathrm{d}y}=-\tfrac{1}{\beta_t^2}\mathbb{E}[x-\alpha_t Y\,|\,\bar{X}_t=x],\qquad(92)$$

$$\nabla\log p_t(x)=\frac{\int_{\mathbb{R}^d}(\alpha_t\nabla\log p_{\text{data}}(y)-(x-\alpha_t y))\mathcal{N}(x;\alpha_t y,\beta_t^2 I)\,p_{\text{data}}(y)\,\mathrm{d}y}{(\alpha_t^2+\beta_t^2)\int_{\mathbb{R}^d}\mathcal{N}(x;\alpha_t y,\beta_t^2 I)\,p_{\text{data}}(y)\,\mathrm{d}y}\qquad(93)$$

$$=\tfrac{1}{\alpha_t^2+\beta_t^2}\mathbb{E}[\alpha_t\nabla\log p_{\text{data}}(Y)-(x-\alpha_t Y)\,|\,\bar{X}_t=x],\qquad(94)$$

The expressions for the Conditional Score Matching loss $\mathcal{L}_{\text{CSM}}$ and the Novel Score Matching loss $\mathcal{L}_{\text{NSM}}$ follow from an argument analogous to equation (90), the only differences being that in the second equality we use equations (92) and (94) instead.

### B.3 Proof of Prop. 3.2: bias-variance decomposition for Adjoint Matching and Sampling

For Adjoint Matching and Sampling, observe that $u(x,t)=\frac{1}{\sqrt{2\eta_t}}\big(v_{\text{ft}}(x,t)-v_{\text{base}}(x,t)\big)$. Hence,

$$\mathbb{E}[\mathcal{L}_{\text{Adj-Match}}(u;X^{\bar{u}})]$$

$$=\mathbb{E}\Big[\tfrac{1}{2}\int_0^1\big\|u(X_t^{\bar{u}},t)+\sigma(t)^\top\tilde{a}(t;X^{\bar{u}})\big\|^2\,\mathrm{d}t\Big]$$

$$=\mathbb{E}\Big[\tfrac{1}{2}\int_0^1\big\|\tfrac{1}{\sqrt{2\eta_t}}\big(v_{\text{ft}}(X_t^{\bar{u}},t)-v_{\text{base}}(X_t^{\bar{u}},t)\big)+\sqrt{2\eta_t}\tilde{a}(t;X^{\bar{u}})\big\|^2\,\mathrm{d}t\Big]$$

$$\overset{(i)}{=}\mathbb{E}\Big[\tfrac{1}{2}\int_0^1\big\|v_{\text{ft}}(X_t^{\bar{u}},t)-v_{\text{base}}(X_t^{\bar{u}},t)+2\eta_t\tilde{a}(t;X^{\bar{u}})\big\|^2\tfrac{1}{2\eta_t}\,\mathrm{d}t\Big]$$

$$=\mathbb{E}\Big[\tfrac{1}{2}\int_0^1\big\|v_{\text{ft}}(X_t^{\bar{u}},t)-\mathbb{E}\big[v_{\text{base}}(X_t^{\bar{u}},t)-2\eta_t\tilde{a}(t;X^{\bar{u}})|X_t^{\bar{u}}\big]\big\|^2\tfrac{1}{2\eta_t}\,\mathrm{d}t\Big]$$

$$\quad+\mathbb{E}\Big[\tfrac{1}{2}\int_0^1\big\|\mathbb{E}\big[v_{\text{base}}(X_t^{\bar{u}},t)-2\eta_t\tilde{a}(t;X^{\bar{u}})|X_t^{\bar{u}}\big]-\big(v_{\text{base}}(X_t^{\bar{u}},t)-2\eta_t\tilde{a}(t;X^{\bar{u}})\big)\big\|^2\tfrac{1}{2\eta_t}\,\mathrm{d}t\Big]$$

$$(95)$$

Observe that equality (i) yields an expression on the same form as equation (24), with $\xi(t,X^{\bar{u}})=v_{\text{base}}(X_t^{\bar{u}},t)+2\eta_t\tilde{a}(t;X^{\bar{u}})$. The second term in the right-hand side of (95) (the variance term) can be simplified to

$$\mathbb{E}\Big[\tfrac{1}{2}\int_0^1\big\|\mathbb{E}\big[2\eta_t\tilde{a}(t;X^{\bar{u}})|X_t^{\bar{u}}\big]-2\eta_t\tilde{a}(t;X^{\bar{u}})\big\|^2\tfrac{1}{2\eta_t}\,\mathrm{d}t\Big]=\mathbb{E}\Big[\int_0^1\eta_t\big\|\mathbb{E}\big[\tilde{a}(t;X^{\bar{u}})|X_t^{\bar{u}}\big]-\tilde{a}(t;X^{\bar{u}})\big\|^2\,\mathrm{d}t\Big]$$

$$\leq\mathbb{E}\Big[\int_0^1\eta_t\big\|\tilde{a}(t;X^{\bar{u}})\big\|^2\,\mathrm{d}t\Big]\leq\int_0^1\eta_t\exp\big(2\int_t^1\chi_s\,\mathrm{d}s\big)\,\mathrm{d}t\times\mathbb{E}\big[\|\nabla_x r(X_1^u)\|^2\big].$$

$$(96)$$

For the particular case in which $p_{\text{data}}$ is Gaussian, we can similarly obtain an equality:

$$\mathbb{E}\Big[\int_0^1\big\|\mathbb{E}\big[2\eta_t\tilde{a}(t;X^{\bar{u}})|X_t^{\bar{u}}\big]-2\eta_t\tilde{a}(t;X^{\bar{u}})\big\|^2\tfrac{1}{2\eta_t}\,\mathrm{d}t\Big]\qquad(97)$$

$$=\mathbb{E}\Big[\int_0^1\eta_t\exp\big(2\int_t^1\chi_s\,\mathrm{d}s\big)\big(\|\nabla_x r(X_1^u)\|^2-\mathbb{E}\big[\|\nabla_x r(X_1^u)\|^2|X_t^u\big]\big)\,\mathrm{d}t\Big],\qquad(98)$$

where the last equality holds by equation (76). Next, we compute $\int_0^1\eta_t\exp\big(2\int_t^1\chi_s\,\mathrm{d}s\big)\,\mathrm{d}t$ in the three subcases:

(i) Föllmer process:

$$\int_0^1 \eta_t \exp\left(2\int_t^1 \chi_s \, ds\right) dt = \int_0^1 \left(\frac{\sigma_1^2}{(1-\bar{\alpha}_t)\sigma_0^2 + \bar{\alpha}_t\sigma_1^2}\right)^2 \frac{\dot{\bar{\alpha}}_t \sigma_0^2}{2} \, dt = \int_0^1 \left(\frac{\sigma_1^2}{(1-t)\sigma_0^2 + t\sigma_1^2}\right)^2 \frac{\sigma_0^2}{2} \, dt = \frac{\sigma_1^2}{2}.$$
(99)

(ii) DDIM/DDPM:

$$\int_0^1 \eta_t \exp\left(2\int_t^1 \chi_s \, ds\right) dt = \int_0^1 \left(\frac{\sigma_1^2\sqrt{\bar{\alpha}_t}}{(1-\bar{\alpha}_t)\sigma_0^2 + \bar{\alpha}_t\sigma_1^2}\right)^2 \frac{\dot{\bar{\alpha}}_t \sigma_0^2}{2\bar{\alpha}_t} \, dt = \int_0^1 \left(\frac{\sigma_1^2\sqrt{t}}{(1-t)\sigma_0^2 + t\sigma_1^2}\right)^2 \frac{\sigma_0^2}{2t} \, dt = \frac{\sigma_1^2}{2}.$$
(100)

(iii) Rectified Flow:

$$\int_0^1 \eta_t \exp\left(2\int_t^1 \chi_s \, ds\right) dt = \int_0^1 \left(\frac{\sigma_1^2\bar{\alpha}_t}{(1-\bar{\alpha}_t)^2\sigma_0^2 + \bar{\alpha}_t^2\sigma_1^2}\right)^2 \frac{(1-\bar{\alpha}_t)\dot{\bar{\alpha}}_t\sigma_0^2}{\bar{\alpha}_t} \, dt$$
$$= \int_0^1 \frac{(1-t)\sigma_0^2}{t}\left(\frac{\sigma_1^2 t}{(1-t)^2\sigma_0^2 + t^2\sigma_1^2}\right)^2 dt = \frac{\sigma_1^2}{2}.$$
(101)

### B.4 Proof of Prop. 3.3: bias-variance decomposition for score matching algorithms

**Target Score Matching** Next, we write the Target Score Matching and Novel Score Matching losses in the general form (24). Plugging $\sigma(t) = \sqrt{2\eta_t}$, we can write

$$v_{\text{ft}}(x,t) = \kappa_t x + 2\eta_t \hat{s}(x,t) \implies \hat{s}(x,t) = \frac{v_{\text{ft}}(x,t) - \kappa_t x}{2\eta_t}.$$
(102)

Thus, for Target Score Matching we have that

$$\mathbb{E}_{Y\sim p_{\text{data}}, \bar{X}_t = \alpha_t Y + \beta_t \varepsilon}\left[\frac{1}{2}\int_0^1 \|\hat{s}(\bar{X}_t, t) - \frac{1}{\alpha_t}\left(\nabla\log p_{\text{base}}(Y) + \nabla r(Y)\right)\|^2 (2\eta_t) \, dt\right]$$

$$= \mathbb{E}_{Y\sim p_{\text{data}}, \bar{X}_t = \alpha_t Y + \beta_t \varepsilon}\left[\frac{1}{2}\int_0^1 \|\frac{v_{\text{ft}}(\bar{X}_t, t) - \kappa_t \bar{X}_t}{2\eta_t} - \frac{1}{\alpha_t}\left(\nabla\log p_{\text{base}}(Y) + \nabla r(Y)\right)\|^2 (2\eta_t) \, dt\right]$$

$$= \mathbb{E}_{Y\sim p_{\text{data}}, \bar{X}_t = \alpha_t Y + \beta_t \varepsilon}\left[\frac{1}{2}\int_0^1 \|v_{\text{ft}}(\bar{X}_t, t) - \kappa_t \bar{X}_t - \frac{2\eta_t}{\alpha_t}\left(\nabla\log p_{\text{base}}(Y) + \nabla r(Y)\right)\|^2 \frac{1}{2\eta_t} \, dt\right]$$

$$= \mathbb{E}_{Y\sim p_{\text{data}}, \bar{X}_t = \alpha_t Y + \beta_t \varepsilon}\left[\frac{1}{2}\int_0^1 \|v_{\text{ft}}(\bar{X}_t, t) - \mathbb{E}\left[\kappa_t \bar{X}_t + \frac{2\eta_t}{\alpha_t}\left(\nabla\log p_{\text{base}}(Y) + \nabla r(Y)\right)|\bar{X}_t\right]\|^2 \frac{1}{2\eta_t} \, dt\right]$$

$$+ \mathbb{E}_{Y\sim p_{\text{data}}, \bar{X}_t = \alpha_t Y + \beta_t \varepsilon}\left[\frac{1}{2}\int_0^1 \|\mathbb{E}\left[\kappa_t \bar{X}_t + \frac{2\eta_t}{\alpha_t}\left(\nabla\log p_{\text{base}}(Y) + \nabla r(Y)\right)|\bar{X}_t\right]\right.$$
$$\left. - \left(\kappa_t \bar{X}_t + \frac{2\eta_t}{\alpha_t}\left(\nabla\log p_{\text{base}}(Y) + \nabla r(Y)\right)\right)\|^2 \frac{1}{2\eta_t} \, dt\right]$$
(103)

The second term in the right-hand side of (103) (the variance term) can be simplified to

$$\mathbb{E}_{Y\sim p_{\text{data}}, \bar{X}_t = \alpha_t Y + \beta_t \varepsilon}\left[\frac{1}{2}\int_0^1 \|\mathbb{E}\left[\frac{2\eta_t}{\alpha_t}\left(\nabla\log p_{\text{base}}(Y) + \nabla r(Y)\right)|\bar{X}_t\right]\right.$$
$$\left. - \frac{2\eta_t}{\alpha_t}\left(\nabla\log p_{\text{base}}(Y) + \nabla r(Y)\right)\|^2 \frac{1}{2\eta_t} \, dt\right]$$

$$= \mathbb{E}_{Y\sim p_{\text{data}}, \bar{X}_t = \alpha_t Y + \beta_t \varepsilon}\left[\frac{1}{2}\int_0^1 \frac{2\eta_t}{\alpha_t^2}\|\mathbb{E}\left[\nabla\log p_{\text{base}}(Y) + \nabla r(Y)|\bar{X}_t\right]\right.$$
$$\left. - \left(\nabla\log p_{\text{base}}(Y) + \nabla r(Y)\right)\|^2 \, dt\right]$$
(104)

$$\leq \int_0^1 \frac{\eta_t}{\alpha_t^2} \, dt \times \mathbb{E}_{Y\sim p_{\text{data}}}\left[\|\nabla\log p_{\text{base}}(Y) + \nabla r(Y)\|^2\right].$$

Observe that $\frac{\eta_t}{\alpha_t^2} = \frac{\beta_t\left(\frac{\dot{\alpha}_t}{\alpha_t}\beta_t - \dot{\beta}_t\right)}{\alpha_t^2}$. And in particular, for the three subcases:

(i) Föllmer process:

$$\frac{\eta_t}{\alpha_t^2} = \frac{\dot{\bar{\alpha}}_t\sigma_0^2}{2\bar{\alpha}_t^2} \implies \int_0^1 \frac{\eta_t}{\alpha_t^2} \, dt = \int_0^1 \frac{\dot{\bar{\alpha}}_t\sigma_0^2}{2\bar{\alpha}_t^2} \, dt = \int_0^1 \frac{\sigma_0^2}{2t^2} \, dt = +\infty.$$
(105)

(ii) DDIM/DDPM:

$$\frac{\eta_t}{\alpha_t^2} = \frac{\frac{\dot{\bar{\alpha}}_t\sigma_0^2}{2\bar{\alpha}_t}}{\bar{\alpha}_t} = \frac{\dot{\bar{\alpha}}_t\sigma_0^2}{2\bar{\alpha}_t^2} \implies \int_0^1 \frac{\eta_t}{\alpha_t^2} \, dt = \int_0^1 \frac{\dot{\bar{\alpha}}_t\sigma_0^2}{2\bar{\alpha}_t^2} \, dt = +\infty$$
(106)

(iii) Rectified Flow:

$$\frac{\eta_t}{\alpha_t^2} = \frac{\frac{(1-\bar{\alpha}_t)\dot{\bar{\alpha}}_t\sigma_0^2}{\bar{\alpha}_t}}{\bar{\alpha}_t^2} = \frac{(1-\bar{\alpha}_t)\dot{\bar{\alpha}}_t\sigma_0^2}{\bar{\alpha}_t^3} \implies \int_0^1 \frac{\eta_t}{\alpha_t^2} \, dt = \int_0^1 \frac{(1-\bar{\alpha}_t)\dot{\bar{\alpha}}_t\sigma_0^2}{\bar{\alpha}_t^3} \, dt = \int_0^1 \frac{(1-t)\dot{\sigma}_0^2}{t^3} \, dt = +\infty$$
(107)

**Conditional Score Matching**  And for Conditional Score Matching, we have that

$$\mathbb{E}_{Y\sim p_{\text{data}}, \bar{X}_t=\alpha_t Y+\beta_t\varepsilon}\left[\frac{1}{2}\int_0^1 \|\hat{s}(\bar{X}_t,t)+\frac{\bar{X}_t-\alpha_t Y}{\beta_t^2}\|^2(2\eta_t)\,\mathrm{d}t\right]$$

$$= \mathbb{E}_{Y\sim p_{\text{data}}, \bar{X}_t=\alpha_t Y+\beta_t\varepsilon}\left[\int_0^1 \|\frac{v_{\text{ft}}(\bar{X}_t,t)-\kappa_t\bar{X}_t}{2\eta_t}+\frac{\bar{X}_t-\alpha_t Y}{\beta_t^2}\|^2(2\eta_t)\,\mathrm{d}t\right]$$

$$= \mathbb{E}_{Y\sim p_{\text{data}}, \bar{X}_t=\alpha_t Y+\beta_t\varepsilon}\left[\frac{1}{2}\int_0^1 \|v_{\text{ft}}(\bar{X}_t,t)-\kappa_t\bar{X}_t+\frac{2\eta_t(\bar{X}_t-\alpha_t Y)}{\beta_t^2}\|^2\frac{1}{2\eta_t}\,\mathrm{d}t\right]$$

$$= \mathbb{E}_{Y\sim p_{\text{data}}, \bar{X}_t=\alpha_t Y+\beta_t\varepsilon}\left[\frac{1}{2}\int_0^1 \|v_{\text{ft}}(\bar{X}_t,t)-\mathbb{E}\left[\kappa_t\bar{X}_t+\frac{2\eta_t(\bar{X}_t-\alpha_t Y)}{\beta_t^2}\big|\bar{X}_t\right]\|^2\frac{1}{2\eta_t}\,\mathrm{d}t\right]$$

$$+ \mathbb{E}_{Y\sim p_{\text{data}}, \bar{X}_t=\alpha_t Y+\beta_t\varepsilon}\left[\frac{1}{2}\int_0^1 \|\mathbb{E}\left[\kappa_t\bar{X}_t+\frac{2\eta_t(\bar{X}_t-\alpha_t Y)}{\beta_t^2}\big|\bar{X}_t\right]-\left(\kappa_t\bar{X}_t+\frac{2\eta_t(\bar{X}_t-\alpha_t Y)}{\beta_t^2}\right)\|^2\frac{1}{2\eta_t}\,\mathrm{d}t\right]. \tag{108}$$

The second term in the right-hand side of (108) (the variance term) can be simplified to

$$\mathbb{E}_{Y\sim p_{\text{data}}, \bar{X}_t=\alpha_t Y+\beta_t\varepsilon}\left[\frac{1}{2}\int_0^1 \|\mathbb{E}\left[\frac{2\eta_t\left(\bar{X}_t-\alpha_t Y\right)}{\beta_t^2}\big|\bar{X}_t\right]-\frac{2\eta_t\left(\bar{X}_t-\alpha_t Y\right)}{\beta_t^2}\|^2\frac{1}{2\eta_t}\,\mathrm{d}t\right] \tag{109}$$

$$= \mathbb{E}_{Y\sim p_{\text{data}}, \bar{X}_t=\alpha_t Y+\beta_t\varepsilon}\left[\int_0^1 \frac{\eta_t}{\beta_t^4}\|\mathbb{E}\left[\nabla\log p_{\text{base}}(Y)+\nabla r(Y)|\bar{X}_t\right]-\left(\nabla\log p_{\text{base}}(Y)+\nabla r(Y)\right)\|^2\,\mathrm{d}t\right] \tag{110}$$

$$\leq \int_0^1 \frac{\eta_t}{\beta_t^4}\,\mathrm{d}t \times \mathbb{E}_{Y\sim p_{\text{data}}}\left[\|\nabla\log p_{\text{base}}(Y)+\nabla r(Y))\|^2\right]. \tag{111}$$

Observe that $\frac{\eta_t}{\beta_t^4}=\frac{\beta_t\left(\frac{\dot{\alpha}_t}{\alpha_t}\beta_t-\dot{\beta}_t\right)}{\beta_t^4}=\frac{\frac{\dot{\alpha}_t}{\alpha_t}-\frac{\dot{\beta}_t}{\beta_t}}{\beta_t^2}$. And in particular, for the three subcases:

(i) Föllmer process:

$$\frac{\eta_t}{\alpha_t^2}=\frac{\dot{\bar{\alpha}}_t\sigma_0^2}{2\bar{\alpha}_t^2(1-\bar{\alpha}_t)^2\sigma_0^4}=\frac{\dot{\bar{\alpha}}_t}{2\bar{\alpha}_t^2(1-\bar{\alpha}_t)^2\sigma_0^2}$$
$$\implies \int_0^1 \frac{\eta_t}{\alpha_t^2}\,\mathrm{d}t=\int_0^1 \frac{\dot{\bar{\alpha}}_t}{2\bar{\alpha}_t^2(1-\bar{\alpha}_t)^2\sigma_0^2}\,\mathrm{d}t=\int_0^1 \frac{1}{2\sigma_0^2 t^2(1-t)^2}\,\mathrm{d}t=+\infty. \tag{112}$$

(ii) DDIM/DDPM:

$$\frac{\eta_t}{\beta_t^4}=\frac{\frac{\dot{\bar{\alpha}}_t\sigma_0^2}{2\bar{\alpha}_t}}{(1-\bar{\alpha}_t)^2\sigma_0^4}=\frac{\dot{\bar{\alpha}}_t}{2\bar{\alpha}_t(1-\bar{\alpha}_t)^2\sigma_0^2}$$
$$\implies \int_0^1 \frac{\eta_t}{\beta_t^4}\,\mathrm{d}t=\int_0^1 \frac{\dot{\bar{\alpha}}_t}{2\bar{\alpha}_t(1-\bar{\alpha}_t)^2\sigma_0^2}\,\mathrm{d}t=\int_0^1 \frac{1}{2t(1-t)^2\sigma_0^2}\,\mathrm{d}t=+\infty. \tag{113}$$

(iii) Rectified Flow:

$$\frac{\eta_t}{\beta_t^4}=\frac{\frac{(1-\bar{\alpha}_t)\dot{\bar{\alpha}}_t\sigma_0^2}{\bar{\alpha}_t}}{(1-\bar{\alpha}_t)^4}=\frac{(1-\bar{\alpha}_t)\dot{\bar{\alpha}}_t\sigma_0^2}{\bar{\alpha}_t(1-\bar{\alpha}_t)^4}=\frac{\dot{\bar{\alpha}}_t\sigma_0^2}{\bar{\alpha}_t(1-\bar{\alpha}_t)^3}$$
$$\implies \int_0^1 \frac{\eta_t}{\beta_t^4}\,\mathrm{d}t=\int_0^1 \frac{\dot{\bar{\alpha}}_t\sigma_0^2}{\bar{\alpha}_t(1-\bar{\alpha}_t)^3}\,\mathrm{d}t=\int_0^1 \frac{\sigma_0^2}{t(1-t)^3}\,\mathrm{d}t=+\infty. \tag{114}$$

**Novel Score Matching**  And for Novel Score Matching, we have that

$$\mathbb{E}_{Y\sim p_{\text{data}}, \bar{X}_t=\alpha_t Y+\beta_t\varepsilon}\left[\frac{1}{2}\int_0^1 \|\hat{s}(\bar{X}_t,t)-\frac{\alpha_t(\nabla\log p_{\text{base}}(Y)+\nabla r(Y))-(\bar{X}_t-\alpha_t Y)}{\alpha_t^2+\beta_t^2}\|^2(2\eta_t)\,\mathrm{d}t\right]$$

$$= \mathbb{E}_{Y\sim p_{\text{data}}, \bar{X}_t=\alpha_t Y+\beta_t\varepsilon}\left[\frac{1}{2}\int_0^1 \|\frac{v_{\text{ft}}(\bar{X}_t,t)-\kappa_t\bar{X}_t}{2\eta_t}-\frac{\alpha_t(\nabla\log p_{\text{base}}(Y)+\nabla r(Y))-(\bar{X}_t-\alpha_t Y)}{\alpha_t^2+\beta_t^2}\|^2(2\eta_t)\,\mathrm{d}t\right]$$

$$= \mathbb{E}_{Y\sim p_{\text{data}}, \bar{X}_t=\alpha_t Y+\beta_t\varepsilon}\left[\frac{1}{2}\int_0^1 \|v_{\text{ft}}(\bar{X}_t,t)-\kappa_t\bar{X}_t-\frac{2\eta_t\left(\alpha_t(\nabla\log p_{\text{base}}(Y)+\nabla r(Y))-(\bar{X}_t-\alpha_t Y)\right)}{\alpha_t^2+\beta_t^2}\|^2\frac{1}{2\eta_t}\,\mathrm{d}t\right]$$

$$= \mathbb{E}_{Y\sim p_{\text{data}}, \bar{X}_t=\alpha_t Y+\beta_t\varepsilon}\left[\frac{1}{2}\int_0^1 \|v_{\text{ft}}(\bar{X}_t,t)-\mathbb{E}\left[\kappa_t\bar{X}_t+\frac{2\eta_t\left(\alpha_t(\nabla\log p_{\text{base}}(Y)+\nabla r(Y))-(\bar{X}_t-\alpha_t Y)\right)}{\alpha_t^2+\beta_t^2}\big|\bar{X}_t\right]\|^2\frac{1}{2\eta_t}\,\mathrm{d}t\right]$$

$$+ \mathbb{E}_{Y\sim p_{\text{data}}, \bar{X}_t=\alpha_t Y+\beta_t\varepsilon}\left[\frac{1}{2}\int_0^1 \|\mathbb{E}\left[\kappa_t\bar{X}_t+\frac{2\eta_t\left(\alpha_t(\nabla\log p_{\text{base}}(Y)+\nabla r(Y))-(\bar{X}_t-\alpha_t Y)\right)}{\alpha_t^2+\beta_t^2}\big|\bar{X}_t\right]\right.$$

$$\left.-\left(\kappa_t\bar{X}_t+\frac{2\eta_t\left(\alpha_t(\nabla\log p_{\text{base}}(Y)+\nabla r(Y))-(\bar{X}_t-\alpha_t Y)\right)}{\alpha_t^2+\beta_t^2}\right)\|^2\frac{1}{2\eta_t}\,\mathrm{d}t\right]. \tag{115}$$

The second term in the right-hand side of (115) (the variance term) can be simplified to

$$
\begin{aligned}
\mathbb{E}_{Y \sim p_{\text{data}}, \bar{X}_t = \alpha_t Y + \beta_t \varepsilon} & \Big[ \tfrac{1}{2} \int_0^1 \big\| \mathbb{E}\big[ \tfrac{2\eta_t \left( \alpha_t (\nabla \log p_{\text{base}}(Y) + \nabla r(Y)) + \alpha_t Y \right)}{\alpha_t^2 + \beta_t^2} | \bar{X}_t \big] \\
& \qquad - \big( \tfrac{2\eta_t \left( \alpha_t (\nabla \log p_{\text{base}}(Y) + \nabla r(Y)) + \alpha_t Y \right)}{\alpha_t^2 + \beta_t^2} \big) \big\|^2 \tfrac{1}{2\eta_t} \, dt \Big] \\
= \mathbb{E}_{Y \sim p_{\text{data}}, \bar{X}_t = \alpha_t Y + \beta_t \varepsilon} & \Big[ \int_0^1 \tfrac{\eta_t \alpha_t^2}{\alpha_t^2 + \beta_t^2} \big\| \mathbb{E}\big[ \nabla \log p_{\text{base}}(Y) + \nabla r(Y) + Y | \bar{X}_t \big] \\
& \qquad - \big( \nabla \log p_{\text{base}}(Y) + \nabla r(Y) + Y \big) \big\|^2 \, dt \Big] \\
\leq \int_0^1 \tfrac{\eta_t \alpha_t^2}{\alpha_t^2 + \beta_t^2} \, dt & \times \mathbb{E}_{Y \sim p_{\text{data}}} \big[ \| \nabla \log p_{\text{base}}(Y) + \nabla r(Y) + Y \|^2 \big].
\end{aligned}
\tag{116}
$$

And in particular, for the three subcases:

(i) Föllmer process:

$$
\frac{\eta_t \alpha_t^2}{\alpha_t^2 + \beta_t^2} = \frac{\dot{\bar{\alpha}}_t \sigma_0^2}{2} \frac{\bar{\alpha}_t^2}{\bar{\alpha}_t^2 + \bar{\alpha}_t (1 - \bar{\alpha}_t) \sigma_0^2} = \frac{\dot{\bar{\alpha}}_t \sigma_0^2 \bar{\alpha}_t}{2(\bar{\alpha}_t + (1 - \bar{\alpha}_t) \sigma_0^2)} = \frac{\dot{\bar{\alpha}}_t \sigma_0^2 \bar{\alpha}_t}{2(\sigma_0^2 + (1 - \sigma_0^2) \bar{\alpha}_t)}
$$

$$
\implies \int_0^1 \frac{\eta_t \alpha_t^2}{\alpha_t^2 + \beta_t^2} \, dt = \int_0^1 \frac{\dot{\bar{\alpha}}_t \sigma_0^2 \bar{\alpha}_t}{2(\sigma_0^2 + (1 - \sigma_0^2) \bar{\alpha}_t)} \, dt = \int_0^1 \frac{\sigma_0^2 t}{2(\sigma_0^2 + (1 - \sigma_0^2) t)} \, dt
$$

$$
= \begin{cases} \frac{\sigma_0^2}{2(1 - \sigma_0^2)} + \frac{\sigma_0^4}{2(1 - \sigma_0^2)^2} \log(\sigma_0^2) & \text{if } \sigma_0 \neq 1, \\ \frac{1}{4} & \text{if } \sigma_0 = 1. \end{cases}
\tag{117}
$$

(ii) DDIM/DDPM:

$$
\frac{\eta_t \alpha_t^2}{\alpha_t^2 + \beta_t^2} = \frac{\frac{\dot{\bar{\alpha}}_t \sigma_0^2}{2 \bar{\alpha}_t} \bar{\alpha}_t}{\bar{\alpha}_t + (1 - \bar{\alpha}_t) \sigma_0^2} = \frac{\dot{\bar{\alpha}}_t \sigma_0^2}{2(\bar{\alpha}_t + (1 - \bar{\alpha}_t) \sigma_0^2)}
\tag{118}
$$

$$
\implies \int_0^1 \frac{\eta_t \alpha_t^2}{\alpha_t^2 + \beta_t^2} \, dt = \int_0^1 \frac{\dot{\bar{\alpha}}_t \sigma_0^2}{2(\bar{\alpha}_t + (1 - \bar{\alpha}_t) \sigma_0^2)} \, dt = \int_0^1 \frac{\sigma_0^2}{2(t + (1 - t) \sigma_0^2)} \, dt = \begin{cases} -\frac{\sigma_0^2}{2(1 - \sigma_0^2)} \log(\sigma_0^2) & \text{if } \sigma_0 \neq 1, \\ \frac{1}{2} & \text{if } \sigma_0 = 1. \end{cases}
\tag{119}
$$

(iii) Rectified Flow:

$$
\frac{\eta_t \alpha_t^2}{\alpha_t^2 + \beta_t^2} = \frac{\frac{(1 - \bar{\alpha}_t) \dot{\bar{\alpha}}_t \sigma_0^2}{\bar{\alpha}_t} \bar{\alpha}_t^2}{\bar{\alpha}_t^2 + \bar{\beta}_t^2} = \frac{(1 - \bar{\alpha}_t) \dot{\bar{\alpha}}_t \bar{\alpha}_t \sigma_0^2}{\bar{\alpha}_t^2 + \bar{\beta}_t^2}
\tag{120}
$$

$$
\implies \int_0^1 \frac{\eta_t \alpha_t^2}{\alpha_t^2 + \beta_t^2} \, dt = \int_0^1 \frac{(1 - \bar{\alpha}_t) \dot{\bar{\alpha}}_t \bar{\alpha}_t \sigma_0^2}{\bar{\alpha}_t^2 + \bar{\beta}_t^2} \, dt = \int_0^1 \frac{(1 - t) t \sigma_0^2}{t^2 + (1 - t)^2} \, dt = \frac{\sigma_0^2 (\pi - 2)}{4}
\tag{121}
$$

# C PROOFS FOR THE THERMODYNAMICS-BASED METHODS

## C.1 PROOF OF PROP. 4.1: CMCD FOR EXPONENTIAL TILTING

We apply Prop. A.1 with $T = 1$, and the choices $\Gamma_0 = \mathcal{N}(0, \beta_0^2 I)$, $\Gamma_1 = p^{\text{base}}$, $\mu \propto \mathcal{N}(0, \beta_0^2 I) \exp(r_0)$, $\nu \propto p^{\text{base}} \exp(r_1)$, and

$$
\gamma_t^+(x) = b_\sigma(x, t) = \kappa_t x + \big( \tfrac{\sigma(t)^2}{2} + \eta_t \big) \mathfrak{s}_t(x),
\tag{122}
$$

$$
\gamma_t^-(x) = b_\sigma(x, t) - \sigma(t)^2 \mathfrak{s}_t(x) = \kappa_t x + \big( -\tfrac{\sigma(t)^2}{2} + \eta_t \big) \mathfrak{s}_t(x),
\tag{123}
$$

$$
\begin{aligned}
a_t(x) &= \kappa_t x + \big( \tfrac{\sigma(t)^2}{2} + \eta_t \big) \big( \mathfrak{s}_t(x) + \nabla r_t(x) \big) + v(x, t) \\
&= \gamma_t^+(x) + \big( \tfrac{\sigma(t)^2}{2} + \eta_t \big) \nabla r_t(x) + v(x, t),
\end{aligned}
\tag{124}
$$

$$
\begin{aligned}
b_t(x) &= a_t(x) - \sigma(t)^2 \big( \mathfrak{s}_t(x) + \nabla r_t(x) \big) \\
&= \kappa_t x + \big( -\tfrac{\sigma(t)^2}{2} + \eta_t \big) \big( \mathfrak{s}_t(x) + \nabla r_t(x) \big) + v(x, t) \\
&= \gamma_t^-(x) + \big( -\tfrac{\sigma(t)^2}{2} + \eta_t \big) \nabla r_t(x) + v(x, t),
\end{aligned}
\tag{125}
$$

Observe that when $Y$ solves $\overrightarrow{\mathrm{d}Y_t} = a_t(Y_t)\,\mathrm{d}t + \sigma(t)\,\overrightarrow{\mathrm{d}W_t}$, it also solves $\overleftarrow{\mathrm{d}Y_t} = a_t(Y_t)\,\mathrm{d}t + \sigma(t)\,\overleftarrow{\mathrm{d}W_t}$ because $\sigma$ does not depend on the position. When $Y$ solves these SDEs, Prop. A.1 implies that

$$
\begin{aligned}
\log \frac{\mathrm{d}\overrightarrow{\mathbb{P}}^{\mu,a}}{\mathrm{d}\overleftarrow{\mathbb{P}}^{\nu,b}}(Y) = {}& r_0(Y_0) - \log \mathbb{E}_{\mathcal{N}(0,\beta_0^2 \mathrm{I})}[\exp(r_0)] - \big(r_1(Y_1) - \log \mathbb{E}_{p^{\mathrm{base}}}[\exp(r_1)]\big) \\
& + \int_0^1 \sigma(t)^{-2}\big\langle \big(\tfrac{\sigma(t)^2}{2} + \eta_t\big)\nabla r_t(Y_t) + v(Y_t,t), \\
& \qquad\qquad a_t(Y_t)\,\mathrm{d}t + \sigma(t)\,\overrightarrow{\mathrm{d}W_t} - \tfrac{1}{2}\big(a_t + \gamma_t^+\big)(Y_t)\,\mathrm{d}t\big\rangle \\
& - \int_0^1 \sigma(t)^{-2}\big\langle \big(-\tfrac{\sigma(t)^2}{2} + \eta_t\big)\nabla r_t(Y_t) + v(Y_t,t), \\
& \qquad\qquad a_t(Y_t)\,\mathrm{d}t + \sigma(t)\,\overleftarrow{\mathrm{d}W_t} - \tfrac{1}{2}\big(b_t + \gamma_t^-\big)(Y_t)\,\mathrm{d}t\big\rangle \\
= {}& r_0(Y_0) - \log \mathbb{E}_{\mathcal{N}(0,\beta_0^2 \mathrm{I})}[\exp(r_0)] - \big(r_1(Y_1) - \log \mathbb{E}_{p^{\mathrm{base}}}[\exp(r_1)]\big) \qquad (126) \\
& + \int_0^1 \sigma(t)^{-2}\big\langle \big(\tfrac{\sigma(t)^2}{2} + \eta_t\big)\nabla r_t(Y_t) + v(Y_t,t), \\
& \qquad\qquad \tfrac{1}{2}\big(\big(\tfrac{\sigma(t)^2}{2} + \eta_t\big)\nabla r_t(Y_t) + v(Y_t,t)\big)\,\mathrm{d}t + \sigma(t)\,\overrightarrow{\mathrm{d}W_t}\big\rangle \\
& - \int_0^1 \sigma(t)^{-2}\big\langle \big(-\tfrac{\sigma(t)^2}{2} + \eta_t\big)\nabla r_t(Y_t) + v(Y_t,t), \\
& \qquad\qquad \tfrac{1}{2}\big(\big(-\tfrac{\sigma(t)^2}{2} + \eta_t\big)\nabla r_t(Y_t) + v(Y_t,t)\big)\,\mathrm{d}t + \sigma(t)\,\overleftarrow{\mathrm{d}W_t} \\
& \qquad\qquad + \sigma(t)^2\big(\mathfrak{s}_t(Y_t) + \nabla r_t(Y_t)\big)\,\mathrm{d}t\big\rangle,
\end{aligned}
$$

where the last equality holds because

$$
\tfrac{1}{2}\big(a_t(x) - \gamma_t^+(x)\big) = \tfrac{1}{2}\big(\big(\tfrac{\sigma(t)^2}{2} + \eta_t\big)\nabla r_t(x) + v(x,t)\big) \qquad (127)
$$

and

$$
\begin{aligned}
a_t(x) - \tfrac{1}{2}\big(b_t(x) - \gamma_t^-(x)\big) &= \tfrac{1}{2}\big(b_t(x) - \gamma_t^-(x)\big) + \sigma(t)^2\big(\mathfrak{s}_t(x) + \nabla r_t(x)\big) \\
&= \tfrac{1}{2}\big(\big(-\tfrac{\sigma(t)^2}{2} + \eta_t\big)\nabla r_t(x) + v(x,t)\big) + \sigma(t)^2\big(\mathfrak{s}_t(x) + \nabla r_t(x)\big).
\end{aligned} \qquad (128)
$$

The right-hand side of (126) can be further simplified into

$$
\begin{aligned}
\log \frac{\mathrm{d}\overrightarrow{\mathbb{P}}^{\mu,a}}{\mathrm{d}\overleftarrow{\mathbb{P}}^{\nu,b}}(Y) = {}& r_0(Y_0) - \log \mathbb{E}_{\mathcal{N}(0,\beta_0^2 \mathrm{I})}[\exp(r_0)] - \big(r_1(Y_1) - \log \mathbb{E}_{p^{\mathrm{base}}}[\exp(r_1)]\big) \\
& + \int_0^1 \Big[ -\langle v(Y_t,t), \mathfrak{s}_t(Y_t)\rangle + \big(\tfrac{\sigma(t)^2}{2} - \eta_t\big)\langle \nabla r_t(Y_t), \mathfrak{s}_t(Y_t)\rangle + \tfrac{\sigma(t)^2}{2}\|\nabla r_t(Y_t)\|^2 \Big]\,\mathrm{d}t \\
& + \int_0^1 \Big[ \sigma(t)^{-1}\big(\langle v(Y_t,t) + \eta_t \nabla r_t(Y_t), \overrightarrow{\mathrm{d}W_t}\rangle - \langle v(Y_t,t) + \eta_t \nabla r_t(Y_t), \overleftarrow{\mathrm{d}W_t}\rangle\big) \\
& \qquad + \tfrac{\sigma(t)}{2}\big(\langle \nabla r_t(Y_t), \overrightarrow{\mathrm{d}W_t}\rangle + \langle \nabla r_t(Y_t), \overleftarrow{\mathrm{d}W_t}\rangle\big)\Big].
\end{aligned} \qquad (129)
$$

To obtain this simplification, it is convenient to define $w(Y_t,t) := v(Y_t,t) + \eta_t \nabla r_t(Y_t)$ and $w^+(x,t) := w(x,t) + \tfrac{\sigma(t)^2}{2}\nabla r_t(Y_t)$, $w^-(x,t) := w(x,t) - \tfrac{\sigma(t)^2}{2}\nabla r_t(Y_t)$, which means that we can rewrite the right-hand side of (126) as

$$
\begin{aligned}
& \int_0^1 \sigma(t)^{-2}\big\langle w^+, \tfrac{1}{2}w^+\,\mathrm{d}t + \sigma(t)\,\overrightarrow{\mathrm{d}W_t}\big\rangle \\
& - \int_0^1 \sigma(t)^{-2}\big\langle w^-, \tfrac{1}{2}w^-\,\mathrm{d}t + \sigma(t)\,\overleftarrow{\mathrm{d}W_t} + \sigma(t)^2\big(\mathfrak{s}_t(Y_t) + \nabla r_t(Y_t)\big)\,\mathrm{d}t\big\rangle.
\end{aligned} \qquad (130)
$$

Equation (129) then follows from manipulating this expression.

To conclude the proof, we plug equation (129) into the definition of the KL and log-variance CMCD losses:

$$
\begin{aligned}
\mathcal{L}_{\mathrm{KL-CMCD}}(v) = {}& \mathbb{E}_{Y^v \sim \overrightarrow{\mathbb{P}}^{\mu,a^v}}\Big[\log \frac{\mathrm{d}\overrightarrow{\mathbb{P}}^{\mu,a^v}}{\mathrm{d}\overleftarrow{\mathbb{P}}^{\nu,b^v}}(Y^v)\Big] \\
= {}& \mathbb{E}_{Y^v \sim \overrightarrow{\mathbb{P}}^{\mu,a^v}}\Big[ \int_0^1 \big(-\langle v(Y_t^v,t), \mathfrak{s}_t(Y_t^v)\rangle + \big(\tfrac{\sigma(t)^2}{2} - \eta_t\big)\langle \nabla r_t(Y_t^v), \mathfrak{s}_t(Y_t^v)\rangle + \tfrac{\sigma(t)^2}{2}\|\nabla r_t(Y_t^v)\|^2\big)\,\mathrm{d}t \\
& - \int_0^1 \sigma(t)^{-1}\langle v(Y_t^v,t) + (\eta_t - \tfrac{\sigma(t)^2}{2})\nabla r_t(Y_t^v), \overleftarrow{\mathrm{d}W_t}\rangle - r(Y_1^v)\Big] + \mathrm{const.},
\end{aligned}
$$

$$
(131)
$$

where we write $Y^v := Y$ and $a^v := a$ to make the dependency of $a$ on $v$ explicit. And

$$\mathcal{L}_{\text{Var-CMCD}}(v) = \text{Var}_{Y^v \sim \overrightarrow{\mathbb{P}}^{\mu,a^v}}\left[ \log \frac{\mathrm{d}\overrightarrow{\mathbb{P}}^{\mu,a}}{\mathrm{d}\overleftarrow{\mathbb{P}}^{\nu,b}}(Y^v) \right]$$

$$= \text{Var}_{Y^v \sim \overrightarrow{\mathbb{P}}^{\mu,a^v}}\left[ r_0(Y_0^v) - \log \mathbb{E}_{\mathcal{N}(0,\beta_0^2\mathrm{I})}[\exp(r_0)] - \left(r_1(Y_1^v) - \log \mathbb{E}_{p^{\text{base}}}[\exp(r_1)]\right) \right.$$

$$+ \int_0^1 \left[ -\langle v(Y_t^v,t), \mathfrak{s}_t(Y_t^v)\rangle + \left(\frac{\sigma(t)^2}{2} - \eta_t\right)\langle \nabla r_t(Y_t^v), \mathfrak{s}_t(Y_t^v)\rangle + \frac{\sigma(t)^2}{2}\|\nabla r_t(Y_t^v)\|^2 \right] \mathrm{d}t$$

$$+ \int_0^1 \left[ \sigma(t)^{-1}\left(\langle v(Y_t^v,t) + \eta_t \nabla r_t(Y_t^v), \overrightarrow{\mathrm{d}W_t}\rangle - \langle v(Y_t^v,t) + \eta_t \nabla r_t(Y_t^v), \overleftarrow{\mathrm{d}W_t}\rangle\right) \right.$$

$$\left. \left. + \frac{\sigma(t)}{2}\left(\langle \nabla r_t(Y_t^v), \overrightarrow{\mathrm{d}W_t}\rangle + \langle \nabla r_t(Y_t^v), \overleftarrow{\mathrm{d}W_t}\rangle\right) \right] \right]. \tag{132}$$

Observe that the terms $-\log \mathbb{E}_{\mathcal{N}(0,\beta_0^2\mathrm{I})}[\exp(r_0)]$ and $\log \mathbb{E}_{p^{\text{base}}}[\exp(r_1)]$ are unknown constants that can be removed, because they appear inside of the divergence. This yields the final expression of the log-variance CMCD loss.

## C.2 PROOF OF PROP. 4.2: CROOKS FLUCTUATION THEOREM FOR EXPONENTIAL TILTING

We use the notation of App. C.1. We apply Prop. A.1 with the same choices as in App. C.1, but in this case we leave the expression explicitly in terms of $\overrightarrow{\mathrm{d}Y_t}$ and $\overleftarrow{\mathrm{d}Y_t}$, without assuming that $Y_t$ solves any particular SDE. The expression reads:

$$\log \frac{\mathrm{d}\overrightarrow{\mathbb{P}}^{\mu,a}}{\mathrm{d}\overleftarrow{\mathbb{P}}^{\nu,b}}(Y) = r_0(Y_0) - \log \mathbb{E}_{\mathcal{N}(0,\beta_0^2\mathrm{I})}[\exp(r_0)] - \left(r_1(Y_1) - \log \mathbb{E}_{p^{\text{base}}}[\exp(r_1)]\right)$$

$$+ \int_0^1 \sigma(t)^{-2}\langle \left(\tfrac{\sigma(t)^2}{2} + \eta_t\right)\nabla r_t(Y_t) + v(Y_t,t),$$

$$\overrightarrow{\mathrm{d}Y_t} - \left(\kappa_t Y_t + \left(\tfrac{\sigma(t)^2}{2} + \eta_t\right)\left(\mathfrak{s}_t(Y_t) + \tfrac{1}{2}\nabla r_t(Y_t)\right) + \tfrac{1}{2}v(Y_t,t)\right)\mathrm{d}t\rangle$$

$$- \int_0^1 \sigma(t)^{-2}\langle \left(-\tfrac{\sigma(t)^2}{2} + \eta_t\right)\nabla r_t(Y_t) + v(Y_t,t),$$

$$\overleftarrow{\mathrm{d}Y_t} - \left(\kappa_t Y_t + \left(-\tfrac{\sigma(t)^2}{2} + \eta_t\right)\left(\mathfrak{s}_t(Y_t) + \tfrac{1}{2}\nabla r_t(Y_t)\right) + \tfrac{1}{2}v(Y_t,t)\right)\mathrm{d}t\rangle. \tag{133}$$

This can be simplified to:

$$\log \frac{\mathrm{d}\overrightarrow{\mathbb{P}}^{\mu,a}}{\mathrm{d}\overleftarrow{\mathbb{P}}^{\nu,b}}(Y) = r_0(Y_0) - \log \mathbb{E}_{\mathcal{N}(0,\beta_0^2\mathrm{I})}[\exp(r_0)] - \left(r_1(Y_1) - \log \mathbb{E}_{p^{\text{base}}}[\exp(r_1)]\right)$$

$$+ \int_0^1 \sigma(t)^{-2}\langle \eta_t \nabla r_t(Y_t) + v(Y_t,t), \overrightarrow{\mathrm{d}Y_t} - \overleftarrow{\mathrm{d}Y_t}\rangle + \tfrac{1}{2}\int_0^1 \langle \nabla r_t(Y_t), \overrightarrow{\mathrm{d}Y_t} + \overleftarrow{\mathrm{d}Y_t}\rangle$$

$$- \int_0^1 \left(\langle \kappa_t Y_t, \nabla r_t(Y_t)\rangle + \langle v(Y_t,t), \mathfrak{s}_t(Y_t) + \nabla r_t(Y_t)\rangle\right.$$

$$\left. + 2\eta_t\langle \nabla r_t(Y_t), \mathfrak{s}_t(Y_t)\rangle + \eta_t\|\nabla r_t(Y_t)\|^2\right)\mathrm{d}t. \tag{134}$$

Applying (138) and (139), we obtain that

$$\int_0^1 \sigma(t)^{-2}\langle \eta_t \nabla r_t(Y_t) + v(Y_t,t), \overrightarrow{\mathrm{d}Y_t} - \overleftarrow{\mathrm{d}Y_t}\rangle = -\int_0^1 \left(\eta_t \Delta r_t(Y_t) + \nabla \cdot v(Y_t,t)\right)\mathrm{d}t, \tag{135}$$

$$\tfrac{1}{2}\int_0^1 \langle \nabla r_t(Y_t), \overrightarrow{\mathrm{d}Y_t} + \overleftarrow{\mathrm{d}Y_t}\rangle = r_1(Y_1) - r_0(Y_0) - \int_0^1 \partial_t r_t(Y_t)\,\mathrm{d}t, \tag{136}$$

and plugging these into the right-hand side of (137) concludes the proof:

$$\log \frac{\mathrm{d}\overrightarrow{\mathbb{P}}^{\mu,a}}{\mathrm{d}\overleftarrow{\mathbb{P}}^{\nu,b}}(Y) = -\log \mathbb{E}_{\mathcal{N}(0,\beta_0^2\mathrm{I})}[\exp(r_0)] + \log \mathbb{E}_{p^{\text{base}}}[\exp(r_1)]$$

$$- \int_0^1 \left(\langle \kappa_t Y_t + 2\eta_t \mathfrak{s}_t(Y_t), \nabla r_t(Y_t)\rangle + \langle v(Y_t,t), \mathfrak{s}_t(Y_t) + \nabla r_t(Y_t)\rangle + \partial_t r_t(Y_t)\right.$$

$$\left. + \eta_t\|\nabla r_t(Y_t)\|^2 + \eta_t \Delta r_t(Y_t) + \nabla \cdot v(Y_t,t)\right)\mathrm{d}t. \tag{137}$$

**Lemma C.1.** *Suppose that $Y$ satisfies the SDE $\overrightarrow{\mathrm{d}Y_t} = a_t(Y_t)\,\mathrm{d}t + \sigma(t)\,\overrightarrow{\mathrm{d}W_t}$, and that $\omega_t : \mathbb{R}^d \to \mathbb{R}^d$ is differentiable and that $r_t : \mathbb{R}^d \to \mathbb{R}^d$ twice-differentiable with respect to the position variable and differentiable with respect to the time variable. We have that*

$$\int_0^T \langle \omega_t(Y_t), \overleftarrow{\mathrm{d}Y_t} \rangle - \int_0^T \langle \omega_t(Y_t), \overrightarrow{\mathrm{d}Y_t} \rangle = \int_0^T \sigma(t)^2 \, \nabla \cdot \omega_t(Y_t)\,\mathrm{d}t, \tag{138}$$

*and*

$$\int_0^T \langle \nabla r_t(Y_t), \overleftarrow{\mathrm{d}Y_t} \rangle + \int_0^T \langle \nabla r_t(Y_t), \overrightarrow{\mathrm{d}Y_t} \rangle = 2\int_0^T \nabla r_t(Y_t) \circ \mathrm{d}Y_t$$
$$= 2\big(r_T(Y_T) - r_0(Y_0) - \int_0^T \partial_t r_t(Y_t)\,\mathrm{d}t\big), \tag{139}$$

*Proof.* We have that

$$\int_0^T \langle \omega_t(Y_t), \overleftarrow{\mathrm{d}Y_t} \rangle - \int_0^T \langle \omega_t(Y_t), \overrightarrow{\mathrm{d}Y_t} \rangle = [\omega(Y), Y]_T, \tag{140}$$

where $[\omega(Y), Y]_t$ denotes the quadratic variation. Since by Itô's lemma,

$$\overrightarrow{\mathrm{d}}\big(\omega_t(Y_t)\big) = \Big(\partial_t \omega_t(Y_t) + \nabla \omega_t(Y_t)^\top a_t(Y_t) + \tfrac{1}{2}\,\sigma(t)^2\,\Delta \omega_t(Y_t)\Big)\,\mathrm{d}t + \sigma(t)\,\nabla \omega_t(Y_t)^\top\,\overrightarrow{\mathrm{d}W_t}, \tag{141}$$

we have that

$$[\omega(Y), Y]_T = \int_0^T \sigma(t)^2\,\nabla \cdot \omega_t(Y_t)\,\mathrm{d}t \tag{142}$$

The first equality in (139) holds by the fact that

$$\int_0^T \langle \nabla r_t(Y_t), \overleftarrow{\mathrm{d}Y_t} \rangle + \int_0^T \langle \nabla r_t(Y_t), \overrightarrow{\mathrm{d}Y_t} \rangle$$
$$= \lim_{|\pi| \to 0} \sum_{k=0}^{K-1} \langle \nabla r_{t_{k+1}}(Y_{t_{k+1}}) - \nabla r_{t_k}(Y_{t_k}), Y_{k+1} - Y_k \rangle := 2\int_0^T \nabla r_t(Y_t) \circ \mathrm{d}Y_t \tag{143}$$

where $\pi = (t_k)_{k=0}^K$ with $0 = t_0 < \cdots < t_K = T$ and $|\pi| = \max_{k=0:K-1}|t_{k+1} - t_k|$, and the second equality holds by Itô's lemma in the Stratonovich formulation. $\square$

## C.3    Proof of Prop. 4.4: NETS for exponential tilting

We use the notation of App. C.1 and App. C.2. Define

$$F_t = \log \mathbb{E}_{x \sim p_1^{\mathrm{base}}}[\exp(r_t(x))]. \tag{144}$$

Since $\log \mathbb{E}_{p^{\mathrm{base}}}[\exp(r_1)] - \log \mathbb{E}_{\mathcal{N}(0, \beta_0^2 \mathrm{I})}[\exp(r_0)] = F_1 - F_0 = \int_0^1 \partial_t F_t\,\mathrm{d}t$, we can rewrite equation (137) as

$$\log \frac{\mathrm{d}\overrightarrow{\mathbb{P}}^{\mu,a}}{\mathrm{d}\overleftarrow{\mathbb{P}}^{\nu,b}}(Y) = -\int_0^1 \Big( \langle \kappa_t Y_t + 2\eta_t \mathfrak{s}_t(Y_t), \nabla r_t(Y_t) \rangle + \langle v(Y_t, t), \mathfrak{s}_t(Y_t) + \nabla r_t(Y_t) \rangle + \partial_t r_t(Y_t)$$
$$+ \eta_t \|\nabla r_t(Y_t)\|^2 + \eta_t\,\Delta r_t(Y_t) + \nabla \cdot v(Y_t, t) - \partial_t F_t \Big)\,\mathrm{d}t. \tag{145}$$

Thus, for an arbitrary process $Y$, using Jensen's inequality

$$\mathbb{E}\big[\log \frac{\mathrm{d}\overrightarrow{\mathbb{P}}^{\mu,a}}{\mathrm{d}\overleftarrow{\mathbb{P}}^{\nu,b}}(Y)^2\big] = \mathbb{E}\Big[\Big( \int_0^1 \Big(\langle \kappa_t Y_t + 2\eta_t \mathfrak{s}_t(Y_t), \nabla r_t(Y_t) \rangle + \langle v(Y_t, t), \mathfrak{s}_t(Y_t) + \nabla r_t(Y_t) \rangle$$
$$+ \partial_t r_t(Y_t) + \eta_t \|\nabla r_t(Y_t)\|^2 + \eta_t\,\Delta r_t(Y_t) + \nabla \cdot v(Y_t, t) - \partial_t F_t \Big)\,\mathrm{d}t\Big)^2 \Big]$$
$$\leq \mathbb{E}\Big[ \int_0^1 \Big( \langle \kappa_t Y_t + 2\eta_t \mathfrak{s}_t(Y_t), \nabla r_t(Y_t) \rangle + \langle v(Y_t, t), \mathfrak{s}_t(Y_t) + \nabla r_t(Y_t) \rangle$$
$$+ \partial_t r_t(Y_t) + \eta_t \|\nabla r_t(Y_t)\|^2 + \eta_t\,\Delta r_t(Y_t) + \nabla \cdot v(Y_t, t) - \partial_t F_t \Big)^2\,\mathrm{d}t \Big], \tag{146}$$

and the right-hand side is the PINN (physics informed neural network) NETS loss for exponential tilting.

We provide an alternative derivation. The Fokker-Planck equation corresponding to the process $X^v$ defined in (13) is:

$$\partial_t \rho_t + \nabla \cdot ((b_\sigma + v)\rho_t) = \tfrac{\sigma^2}{2} \nabla \cdot \nabla \rho_t \tag{147}$$

We rewrite $b_\sigma^r$ as follows:

$$\begin{aligned}
b_\sigma^r(x,t) &= \kappa_t x + \left(\tfrac{\sigma(t)^2}{2} + \eta_t\right)\left(\mathfrak{s}_t(x) + \nabla r_t(x)\right) \\
&= \kappa_t x + \eta_t\left(\mathfrak{s}_t(x) + \nabla r_t(x)\right) + \tfrac{\sigma(t)^2}{2}\left(\mathfrak{s}_t(x) + \nabla r_t(x)\right) \\
&=: \tilde{b}^r(x,t) + \tfrac{\sigma(t)^2}{2}\left(\mathfrak{s}_t(x) + \nabla r_t(x)\right).
\end{aligned} \tag{148}$$

Thus, equation (147) becomes

$$\partial_t \rho_t + \nabla \cdot ((\tilde{b}^r + v)\rho_t) = \tfrac{\sigma^2}{2} \nabla \cdot \left((-\mathfrak{s}_t - \nabla r_t)\rho_t + \nabla \rho_t\right) \tag{149}$$

Now we want $\rho_t^\star \propto \rho_t^{\text{base}} \exp(r_t)$ to fulfill this PDE. We write explicitly

$$\rho_t^\star(x) = \tfrac{\rho_t^{\text{base}}(x)\exp(r_t(x))}{F_t}, \qquad \hat{F}_t = \mathbb{E}_{y \sim \rho_t^{\text{base}}}[\exp(r_t(y))]. \tag{150}$$

Since $\nabla \cdot \left((-\mathfrak{s}_t - \nabla r_t)\rho_t^\star + \nabla \rho_t^\star\right) = 0$ by construction, we must enforce

$$\begin{aligned}
0 &= \partial_t \rho_t^\star + \nabla \cdot ((\tilde{b}^r + v)\rho_t^\star) \\
&= \partial_t\left(\tfrac{\rho_t^{\text{base}}(x)\exp(r_t(x))}{\hat{F}_t}\right) + \nabla \cdot (\tilde{b}^r + v)\rho_t^\star + \langle \tilde{b}^r + v, \nabla \log \rho_t^\star \rangle \rho_t^\star \\
&= \partial_t\left(\log \rho_t^{\text{base}} + r_t - \log \hat{F}_t\right)\rho_t^\star + \nabla \cdot (\tilde{b}^r + v)\rho_t^\star + \langle \tilde{b}^r + v, \mathfrak{s}_t + \nabla r_t \rangle \rho_t^\star.
\end{aligned} \tag{151}$$

Since $\rho_t^{\text{base}}$ satisfies $0 = \partial_t \rho_t^{\text{base}} + \nabla \cdot \left((\kappa_t x + \eta_t \mathfrak{s}_t)\rho_t^{\text{base}}\right) = \partial_t \rho_t^{\text{base}} + \nabla \cdot \left((\tilde{b}^r - \eta_t \nabla r_t)\rho_t^{\text{base}}\right)$, we obtain that

$$\partial_t \log \rho_t^{\text{base}} = -\nabla \cdot \left(\tilde{b}^r - \eta_t \nabla r_t\right) - \langle \tilde{b}^r - \eta_t \nabla r_t, \mathfrak{s}_t \rangle. \tag{152}$$

Plugging this into the right-hand side of (151) and using that $\log \hat{F}_t = F_t$ with $F_t$ defined in (144) yields

$$\begin{aligned}
0 &= -\nabla \cdot \left(\tilde{b}^r - \eta_t \nabla r_t\right) - \langle \tilde{b}^r - \eta_t \nabla r_t, \mathfrak{s}_t \rangle + \partial_t r_t - \partial_t F_t + \nabla \cdot (\tilde{b}^r + v) + \langle \tilde{b}^r + v, \mathfrak{s}_t + \nabla r_t \rangle \\
&= \nabla \cdot \left(v + \eta_t \nabla r_t\right) + \partial_t r_t - \partial_t F_t + \langle v, \mathfrak{s}_t + \nabla r_t \rangle \\
&\quad - \langle \kappa_t x + \eta_t \mathfrak{s}_t, \mathfrak{s}_t \rangle + \langle \kappa_t x + \eta_t\left(\mathfrak{s}_t + \nabla r_t\right), \mathfrak{s}_t + \nabla r_t \rangle \\
&= \nabla \cdot \left(v + \eta_t \nabla r_t\right) + \partial_t r_t - \partial_t F_t + \langle v, \mathfrak{s}_t + \nabla r_t \rangle \\
&\quad - \langle \kappa_t x + \eta_t \mathfrak{s}_t, \mathfrak{s}_t \rangle + \langle \kappa_t x + \eta_t\left(\mathfrak{s}_t + \nabla r_t\right), \mathfrak{s}_t + \nabla r_t \rangle \\
&= \nabla \cdot \left(v + \eta_t \nabla r_t\right) + \partial_t r_t - \partial_t F_t + \langle v, \mathfrak{s}_t + \nabla r_t \rangle + \langle \kappa_t x + 2\eta_t \mathfrak{s}_t, \nabla r_t \rangle + \eta_t \|\nabla r_t\|^2
\end{aligned} \tag{153}$$

Thus, for $\rho_t^\star$ to satisfy the FPE (147), which means that is the law of the marginal $X_t^v$, we need $v$ to satisfy (153). Observe that the terms in the right-hand side of (153) match one to one the terms in the right-hand side of (146). Hence, the NETS loss can be interpreted as a PINN loss on the residual of (153).

# D ADDITIONAL EXPERIMENTS

## D.1 DERIVATION OF THE GENERATIVE SDE WITH DIFFERENT INITIAL VARIANCE $\sigma_0^2$

Consider Rectified Flow with $\sigma(t) = \gamma \eta_t$ for some $\gamma > 0$, and with two different choices for the initial variance: $\sigma_0^2$ and $\tilde{\sigma}_0^2$. We have that

$$b(x,t) = \kappa_t x + (1+\gamma)\eta_t \mathfrak{s}(x,t) = \tfrac{\dot{\bar{\alpha}}_t}{\bar{\alpha}_t} x + \tfrac{(1+\gamma)(1-\bar{\alpha}_t)\dot{\bar{\alpha}}_t \sigma_0^2}{\bar{\alpha}_t}\mathfrak{s}(x,t), \tag{154}$$

$$\tilde{b}(x,t) = \kappa_t x + (1+\gamma)\tilde{\eta}_t \tilde{\mathfrak{s}}(x,t) = \tfrac{\dot{\bar{\alpha}}_t}{\bar{\alpha}_t} x + \tfrac{(1+\gamma)(1-\bar{\alpha}_t)\dot{\bar{\alpha}}_t \tilde{\sigma}_0^2}{\bar{\alpha}_t}\tilde{\mathfrak{s}}(x,t), \tag{155}$$

where $\mathfrak{s}$ and $\tilde{\mathfrak{s}}$ are the scores corresponding to $\sigma_0^2$ and $\tilde{\sigma}_0^2$, respectively. Next, we apply Corollary A.5 to relate $\mathfrak{s}$ and $\tilde{\mathfrak{s}}$. Defining

$$t(r) = \bar{\alpha}^{-1}\Big(\frac{\sigma_0 \bar{\alpha}_r}{\tilde{\sigma}_0(1-\bar{\alpha}_r)+\sigma_0\bar{\alpha}_r}\Big), \qquad s_r = \frac{\tilde{\sigma}_0(1-\bar{\alpha}_r)+\sigma_0\bar{\alpha}_r}{\sigma_0}, \tag{156}$$

we have that

$$\frac{\bar{\alpha}_r}{(1-\bar{\alpha}_r)\tilde{\sigma}_0} = \frac{\bar{\alpha}_{t(r)}}{(1-\bar{\alpha}_{t(r)})\sigma_0}, \qquad \tilde{\mathfrak{s}}(x,r) = \frac{1}{s_r}\mathfrak{s}(x/s_r, t(r)), \tag{157}$$

and

$$\tilde{b}(x,r) = \frac{\dot{\bar{\alpha}}_r}{\bar{\alpha}_r}x + \frac{(1+\gamma)(1-\bar{\alpha}_r)\dot{\bar{\alpha}}_r\tilde{\sigma}_0^2}{\bar{\alpha}_r s_r}\mathfrak{s}\Big(\frac{x}{s_r},t(r)\Big) = \frac{\dot{\bar{\alpha}}_r}{\bar{\alpha}_r}x + \frac{(1+\gamma)(1-\bar{\alpha}_r)\dot{\bar{\alpha}}_r\tilde{\sigma}_0^2}{\bar{\alpha}_r s_r}\frac{\bar{\alpha}_{t(r)}}{(1-\bar{\alpha}_{t(r)})\dot{\bar{\alpha}}_{t(r)}\sigma_0^2}\Big(v\Big(\frac{x}{s_r},t(r)\Big) - \frac{\dot{\bar{\alpha}}_{t(r)}}{\bar{\alpha}_{t(r)}}\frac{x}{s_r}\Big) \tag{158}$$

$$= \frac{\dot{\bar{\alpha}}_r}{\bar{\alpha}_r}x + \frac{\dot{\bar{\alpha}}_r\tilde{\sigma}_0}{s_r}\frac{1+\gamma}{\dot{\bar{\alpha}}_{t(r)}\sigma_0}\Big(v\Big(\frac{x}{s_r},t(r)\Big) - \frac{\dot{\bar{\alpha}}_{t(r)}}{\bar{\alpha}_{t(r)}}\frac{x}{s_r}\Big) = \frac{\dot{\bar{\alpha}}_r}{\bar{\alpha}_r}x + \frac{\dot{\bar{\alpha}}_r}{\dot{\bar{\alpha}}_{t(r)}}\frac{(1+\gamma)\tilde{\sigma}_0}{\tilde{\sigma}_0(1-\bar{\alpha}_r)+\sigma_0\bar{\alpha}_r}\Big(v\Big(\frac{x}{s_r},t(r)\Big) - \frac{\dot{\bar{\alpha}}_{t(r)}}{\bar{\alpha}_{t(r)}}\frac{x}{s_r}\Big) \tag{159}$$

$$= \frac{\dot{\bar{\alpha}}_r}{\dot{\bar{\alpha}}_{t(r)}}\frac{\tilde{\sigma}_0(1+\gamma)}{\tilde{\sigma}_0(1-\bar{\alpha}_r)+\sigma_0\bar{\alpha}_r}v\Big(\frac{x}{s_r},t(r)\Big) + \dot{\bar{\alpha}}_r\Big(\frac{1}{\bar{\alpha}_r} - \frac{1+\gamma}{\bar{\alpha}_{t(r)}}\frac{\tilde{\sigma}_0\sigma_0}{(\tilde{\sigma}_0(1-\bar{\alpha}_r)+\sigma_0\bar{\alpha}_r)^2}\Big)x \tag{160}$$

$$= \frac{\dot{\bar{\alpha}}_r}{\dot{\bar{\alpha}}_{t(r)}}\frac{\tilde{\sigma}_0(1+\gamma)}{\tilde{\sigma}_0(1-\bar{\alpha}_r)+\sigma_0\bar{\alpha}_r}v\Big(\frac{x}{s_r},t(r)\Big) + \frac{\dot{\bar{\alpha}}_r}{\bar{\alpha}_r}\Big(1 - \frac{\tilde{\sigma}_0(1+\gamma)}{\tilde{\sigma}_0(1-\bar{\alpha}_r)+\sigma_0\bar{\alpha}_r}\Big)x \tag{161}$$

where the second equality holds because $\mathfrak{s}(x,t) = \frac{\bar{\alpha}_t}{(1-\bar{\alpha}_t)\dot{\bar{\alpha}}_t\sigma_0^2}\big(v(x,t) - \frac{\dot{\bar{\alpha}}_t}{\bar{\alpha}_t}x\big)$, the third equality holds because of the first equality in (157), the fourth and fifth equalities hold because of the second equality in (156), and the sixth equality holds because $\frac{\bar{\alpha}_{t(r)}}{\bar{\alpha}_r} = \frac{\sigma_0}{\tilde{\sigma}_0(1-\bar{\alpha}_r)+\sigma_0\bar{\alpha}_r}$, and the seventh equality holds by simplifying. And if we want to run inference with a different noise schedule $\sigma(t)$, we have that

$$\tilde{b}_\sigma(x,r) = \tilde{b}(x,r) + \Big(\frac{\sigma(t)^2}{2} + \tilde{\eta}_t\Big)\tilde{\mathfrak{s}}(x,t). \tag{162}$$

## D.2 ADDITIONAL PLOTS

Fig. 3 is the analog of Fig. 2 with training $\sigma_0^2 = 1.5$.

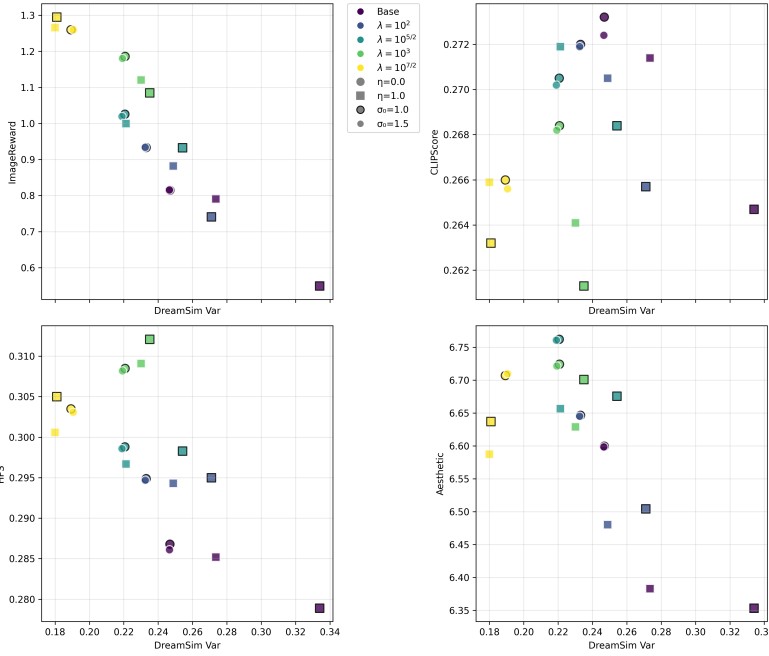

Figure 3: Quality metrics for the base Stable Diffusion 3 model and models fine-tuned at $\sigma_0^2 = 1.5$ and $\lambda \in \{10^2, 10^{5/2}, 10^3\}$, with inference at $\eta \in \{0, 1\}$ and $\sigma_0 = \{1, 1.5\}$.

