# OpenReview forum: "A unified perspective on fine-tuning and sampling with diffusion and flow models"
_ICLR.cc/2026/Conference — ICLR 2026 Conference Withdrawn Submission_

### Official Review · Reviewer_y88r · 2025-10-30

**Soundness:** 2
**Presentation:** 2
**Contribution:** 1
**Rating:** 2
**Confidence:** 3

**Summary:**

The paper studies training diffusion/flow models to sample from exponentially-tilted targets—covering both reward fine-tuning and sampling from unnormalized densities—through two lenses: stochastic optimal control (SOC) and thermodynamics. It (i) gives bounds on the lean adjoint (AM/AS), (ii) derives a common bias–variance view comparing AM/AS with TSM/CSM/NSM (showing variance pathologies for TSM/CSM), (iii) adapts CMCD/NETS with Crooks/Jarzynski identities to the tilting setting, and (iv) reports AM-based text-to-image fine-tuning experiments (SD-1.5/SD-3).

**Strengths:**

1. Lean-adjoint norm bound (Prop. 3.1). Under strong log-concavity of the base, the paper gives an explicit schedule-dependent decay bound on the adjoint’s norm; in the Gaussian case it yields closed-form factors used in AM/AS analysis. This is positioned as theoretical support for AM/AS stability.
2. Unified bias–variance comparison across methods. With a shared weighting w= \eta_t, they show TSM/CSM have infinite variance (blow-ups at t=0/1), while NSM admits finite bounds and AM/AS have a simple finite constant (table summarizing Föllmer, DDIM/DDPM, Rectified-Flow cases). This clarifies when KL-interpretable training is statistically well-posed.
3. Thermodynamics adapted to tilting. The paper derives tilting-aware versions of the controlled Crooks theorem, the escorted Jarzynski equality, and a NETS loss, all incorporating both the base score and reward gradient—tools one could use for thermodynamic training/diagnostics in the tilting formulation.

**Weaknesses:**

1. SOC/unification is known. The exponential-tilting formulation and the memoryless noise-schedule condition enabling SOC (and schedule-agnostic inference) are prior results from Adjoint Matching; here they are restated.
2. Adjoint bound is narrow. The bound hinges on strong log-concavity of the base (or Gaussianity); the paper notes similar behavior is “expected” more generally but does not prove it beyond these assumptions.
3. No new training algorithm; limited empirics. Experiments only apply AM to SD-1.5/SD-3; the thermodynamics adaptations (CMCD/NETS, Crooks/Jarzynski) are not empirically validated here. Overall the work functions as a consolidation + analysis rather than a method contribution.

**Questions:**

Could you please clearly state what is the take-away message of the paper? To me, this submission is a useful consolidation with some clarifying theory, but little in the way of new algorithms or fundamentally new guidance principles.

---

### Official Review · Reviewer_7mmV · 2025-10-31

**Soundness:** 2
**Presentation:** 1
**Contribution:** 2
**Rating:** 2
**Confidence:** 2

**Summary:**

The paper attempts to produce a unified view of several existing diffusion/flow based fine-tuning and sampling methods and analyze some of their properties. More concretely, the paper proposes a disparate set of results
- A bound on the norm of the lean adjoint ODE for Adjoint Matching and Sampling (AM/AS), which potentially provides theoretical support for the empirical performance of these algorithms.
- A bias-variance decomposition for both adjoint-based and score-matching algorithms
- An adaptation of thermodynamic formulations (CMCD, NETS) to the exponential tilting setting.

**Strengths:**

- The attempt to present a wide array of related method in a unified way is laudable. I appreciate Section 2, though it assumes familiarity with a lot of background work.
- Detailed proofs are provided for results. Though, I have not verified all of them in closely.

**Weaknesses:**

- Motivation: At a high level, it is not clear what specific problem the paper is trying to address, why that problem is important and what the key idea is. As written, it seems to be a collection of a set of theoretical results, but without a clear and/or convincing demonstration of their impact on a problem of interest.
- Writing: Lack of clear motivation also makes it hard to follow along and understand the paper, which is already quite dense and assumes familiarity with a significant amount of background knowledge. It is not clear how and which section of the paper/background works is important for the provided results. Several theoretical results are stated, with proof relegated to the appendix, without much discussion of why that result is important and/or what that result unlocks. For example, what are the practical implications of the bound on the lean adjoint ODE? How can this bound be used to guide the design of new algorithms? How sensitive are the results to the choice of the reward model? Beyond speculation, what is the impact of deviation from strong convexity? etc.
- Experimentation: In general, the experimentation section is minimal. However, lack of clear motivation and writing exacerbates the situation by making it difficult to assess what experimentation would be needed, how the presented experiments are sufficient. The experimentation section fails to clearly state what results are being supported by what experiments.


Overall, I think the paper needs to be clearly reorganized and rewritten with a structuring and presentation that makes the answers to aforementioned weaknesses obvious.

**Questions:**

In addition to some of the questions raised above in the weaknesses section, please provide a clear description of what each of the presented theoretical result unlocks or offers beyond what is already known, how that result is valuable and can be used to drive future explorations.

---

### Official Review · Reviewer_9ZXG · 2025-10-31

**Soundness:** 2
**Presentation:** 1
**Contribution:** 2
**Rating:** 2
**Confidence:** 2

**Summary:**

The paper studies the problem of training diffusion/flow models to sample from reward-tilted distributions via fine-tuning or sampling unnormalized densities. The work provides theoretical results to compare recent classes of methods developed to tackle such problems (namely adjoint-based fine-tuning schemes, and score-matching training methods applied over data obtained via inference-time simulation of the tilted distribution). Then, it adapts recent algorithmic schemes based on a thermodynamic interpretation to tackle the exponential tilting problem, and perform experimental evaluations.

**Strengths:**

- The paper aims to provide mathematical understanding on high-relevance problems in diffusion and flow generative modeling. In particular, I believe that theoretical analysis regarding comparison between score-based methods for reward fine-tuning and adjoint-based methods is highly valuable.

- Sec. 2 and 3 provide a fairly interesting unifying lens regarding the problem of computing a diffusion/flow model inducing a distribution matching a reward-tilted distribution.

- The paper seems to provide a potentially interesting mathematical and algorithmic viewpoint on the aforementioned problem (but unfortunately not sufficiently well structured and presented, as discussed in the following)

**Weaknesses:**

- (main concern 1, presentation) On a high level, the paper is written extremely poorly (given the target audience). While I might expect other fields to appreciated un-motivated purely mathematical results, computer science / machine learning entails to consider computational and algorithmic aspects that are deeply neglected in the presentation of this work, leading to profound confusion of its exposition. I will try to mention a subset of examples in the following list:
  1. (abstract, first line) "training diffusion models" ... "subsumes sampling..." and reward "fine-tuning...". Crucially, training/fine-tuning/sampling are different algorithmic problems and training does not subsume the others. Training refers to learning a model from data, fine-tuning to adapting an already available model. Sampling, in this context, typically refers to inference-time adaptation of the diffusion/flow process to sample (typically) from the tilted distribution, while here I believe it is used to indicate classic sampling of unnormalized density. Crucially, sampling as intended here does not even seem to involve data. The same confusion arises in multiple parts of the paper (e.g., line 54-56)
  2. (Sec. 2.2) The exponential tilting problem is defined as the task of modifying the model (line 117) such that it samples from the tilted. Then, 2 main settings are presented: (i) reward fine-tuning and (ii) sampling. Crucially, (ii) does not require an initial model, as the optimal density p^* does not depend on existing data or pre-trained model (see concern below). As a consequence, this problem setting seems wrong.
   3. Sec. 4 aims to introduce an Algorithm, but effectively it 'adapts' existing algorithms that do not seem to be introduced at all. Concretely, it is customary to clarify input/outputs of algorithms, typically provide a pseudocode etc. Presenting mathematical results useful for algorithm design is not sufficient to claim that an algorithm is presented. In particular in this case, since the implicitly proposed scheme would be an adaptation of existing (not presented) algorithms, it is even more essential to clarify the proposed method.

- (main concern 2, lack of motivations/clarity) Most contributions of the work are not properly motivated or explained sufficiently clearly. Some examples:
  1. The contributions listed at the end of the abstract are very poorly connected with the introduced logic.
  2. Same holds for the contributions list within the Introduction
  3. The paper mentions multiple times the thermodynamics framework and algorithm of Vargas et al., (e.g. line 62) but it seems to me that these are never sufficiently well presented, rendering the whole work very hard to follow properly.
  4. Sec. 3 should start introducing why that list of methods is presented....
  5. line 193, exactly why one would wish to bound the norm is not explained. Similarly, the presented bounds should be further discussed.

- (main concern 3, problem setting) The presented 'thermodynamics' formulation is poorly presented, and therefore quite unclear. In particular, the sec. 'The thermodynamics formulation' (line 148) starts with 'Methods like X and Y were developed in a setting where...'. Crucially, it seems to me that lines 151-155 aim to reduce the tilted-distribution sampling problem to a specific sub-case of this setting, but effectively only reduce an irrelevant instance of this problem (e.g. when p_base is data-independent). It might also be the case that the authors were only trying to clarify the difference between the settings. Unfortunately this is unclear due to poor writing, but in any case, what exactly 'the thermodynamics formulation' is (and why thermodynamics?) is unclear to me.

- (main concern 4, method/experiments) Sec. 4 claims to introduce an algorithm by its title. This would arguably be the most practically relevant contribution of this work, which would be otherwise presenting only theoretical results (also fine, but requires a different evaluation). Besides being not explicitly presented, the algorithm does not seem to be even experimentally evaluated at all.


Given the very poor/imprecise writing (for the average audience of this conference), weakly-motivated theoretical results, and unclear and not evaluated algorithm, my current score is negative. Nonetheless, I believe this work might contain interesting ideas that would require further practical development (or more clear theoretical motivations and discussions) as well as a significant rewriting.

**Questions:**

Given convincing clarifications of the points mentioned within the Weaknesses sec. of my review I would be happy to increase my score. In particular, let me know if I misinterpreted/misunderstood any of the points (this is indeed possible due to uncommon writing from a generative modeling viewpoint, or at least the one of my sub-fields of expertise)

---

### Official Review · Reviewer_xkXU · 2025-11-01

**Soundness:** 3
**Presentation:** 2
**Contribution:** 2
**Rating:** 2
**Confidence:** 3

**Summary:**

This paper discusses  fine-tuning and sampling for diffusion/flow models by framing both as sampling from an exponentially tilted target, covering reward fine-tuning and unnormalized densities. It analyzes adjoint-based methods via a new norm bound on the lean adjoint and gives bias–variance decompositions that compare adjoint and score-matching losses, highlighting advantages for AM/AS and NSM. The authors conduct experiments with SD1.5 and 3 with AM.

**Strengths:**

I like the theoretical perspective of the paper, i.e. Prop 3.1 is novel and the variance comparison between many existing works are quite useful for the community (Table 1).

**Weaknesses:**

I feel like the current stage of the paper is incomplete especially given the lack of experiments related to the theory discovered  and not a coherent story. I made some suggestions in the question section which hopefully will be helpful.

**Questions:**

Proposition 3.1 is useful for demonstrating that the lean adjoint objective is more stable.  I think it is much more interesting and potentially needed if the author compares the norm of the lean adjoint objective to the original adjoint objective.  Because then we can have a well-aligned empirical observation with the matching theoretical justification. As the AM author mentions, the adjoint objective is not stable to train.

I highly suggest that the authors elaborate more on the theory discussed in the paper, when I read the paper, I felt like there was not enough discussion on the importance of them, how are they relevant to the fine-tuning task etc, which makes the paper sound like less motivated. And I'm not familiar with the context in section 4, but I feel section 4 is disconnected to the rest of the AM stories.

In the introduction section, the authors claim that this paper "refines the techniques of Domingo-Enrich et al. (2025)." What exactly did this paper refine?

---

### Note · Authors · 2025-11-13

**Comment:**

We thank the reviewers, and agree with several of the points raised, which will be useful for us to improve our work. Given the reviews, we acknowledge that ICLR may not be the best-suited venue for this paper.

**Withdrawal Confirmation:**

I have read and agree with the venue's withdrawal policy on behalf of myself and my co-authors.